# Towards Feedback-to-Plan Decisions
# for Self-Evolving LLM Agents in CUDA Kernel Generation

**Yee Hin Chong** [1]  **Jiaming Wu** [1]  **Youhui Zhang** [1 2]  **Peng Qu** [1 2]

## Abstract

Large language models (LLMs) have shown strong empirical gains as self-evolving agents for CUDA kernel generation, driven by feedback-conditioned planning across generations. However, how planning decisions attribute and combine heterogeneous feedback signals remains opaque. Standard end-to-end ablations fail to resolve this question, as iterative planning amplifies early perturbations and conflates feedback effects with trajectory-dependent drift.

We introduce `CUDAnalyst`, a unified analysis layer for controlled, generation-level attribution of planning decisions to feedback components via trajectory freezing and selective feedback injection. `CUDAnalyst` enables stable generation-level evaluation and principled coalitional-style attribution of feedback effects and interactions. Our results show that explicit planning is beneficial only when feedback is aligned, that effective planning emerges from structured multi-feedback interactions, and that high-level plans from stronger reasoning models can partially transfer to weaker ones. These trends hold across reference backbones, representative workloads, and reference induction regimes, indicating that the identified feedback-to-plan structure is robust within the controlled axes studied.

Code: https://github.com/yuxuan-z19/cudanalyst

## 1. Introduction

Large language models (LLMs) are increasingly deployed as *self-evolving agents* for CUDA kernel generation, where

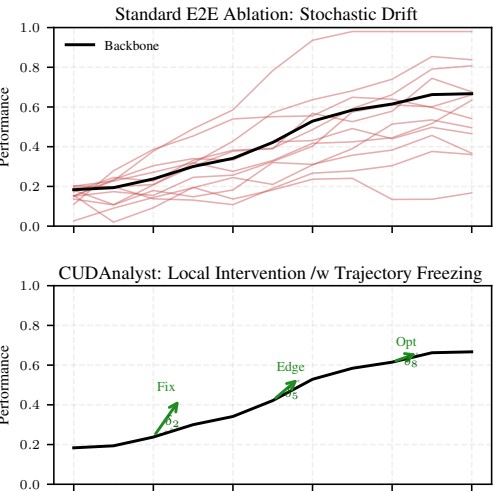

Figure 1. Comparison of end-to-end ablation and intervention on frozen trajectory. E2E suffers from trajectory drift, thus it is unable to present precise causal attribution.

programs are iteratively refined through feedback-driven planning across generations (Zhang et al., 2026b; Wei et al., 2025; Kong et al., 2026). In these systems, planning serves as an explicit decision function that translates heterogeneous diagnostic feedback, ranging from static analyses to runtime measurements, into concrete code modification plans. Despite growing empirical success, **how individual feedback components shape feedback-to-plan decisions at the generation level remains poorly understood**. Lacking utility-aware analysis, practitioners often aggregate diagnostics indiscriminately, obscuring which feedback components meaningfully influence the current planning decisions and, in turn, precluding principled agent design.

Most existing evaluations rely on *end-to-end ablation*, restarting the evolutionary process, either from scratch or from a checkpoint, after modifying a feedback component and reporting only outcomes (Novikov et al., 2025; Zhang et al., 2026a; Liu et al., 2024b). Such protocols are ill-suited for self-evolving LLM agents: iterative planning amplifies early perturbations, making feedback effects inseparable from trajectory-specific drift (Fig. 1). Moreover, aggregating non-monotonic, generation-level outcomes into a single

[1]Department of Computer Science and Technology, Tsinghua University, Beijing, China [2]Beijing National Research Center for Information Science and Technology, Beijing, China. Correspondence to: Peng Qu <qp2018@mail.tsinghua.edu.cn>.

*Proceedings of the 43rd International Conference on Machine Learning*, Seoul, South Korea. PMLR 306, 2026. Copyright 2026 by the author(s).

scalar obscures when specific feedback signals matter and how they interact. As a result, **end-to-end ablation conflates feedback effects with trajectory-dependent drift**, limiting its usefulness for analyzing feedback-to-plan decisions.

To address this limitation, we introduce `CUDAnalyst`, a unified analysis layer built on a simple insight: *feedback attribution must be performed at fixed generations to avoid confounding from cross-generation drift*. By freezing intermediate program states before planning, `CUDAnalyst` enables controlled interventions on feedback signals and supports principled coalitional-style attribution of their contributions and interactions.

Using these capabilities, we systematically evaluate how feedback components shape planning decisions and uncover four main findings that hold consistently across generation backbones, representative workloads and reference induction regimes:

1. Explicit planning is effective only when grounded in feedback, with feedback-aligned planning yielding stable generation-level improvements.

2. Planning effectiveness arises from interactions among multiple feedback components, reflecting stable dependencies on joint feedback availability.

3. Feedback summarization facilitates but does not replace explicit planning, particularly benefiting weaker models.

4. Plans generated by stronger models partially transfer to weaker models within the same model family.

## 2. Related Work

Recent work on LLM-driven CUDA kernel generation increasingly adopts *self-evolving agent* frameworks, replacing one-shot synthesis with iterative refinement guided by feedback-to-plan loops (Li et al., 2025a; Dong et al., 2026; Tschand et al., 2025). These agents incorporate heterogeneous feedback, such as debugging information, static analyses, and runtime measurements, along with retrieved references, into planning contexts to drive successive kernel revisions and achieve substantial performance gains. A concise summary of representative approaches is provided in App. A. While self-evolving agents vary across domains, CUDA kernel generation predominantly relies on feedback-driven evolutionary scaffolds due to offline compilation and execution constraints; our study focuses on this setting.

Despite recent advances in self-evolving agents for CUDA kernel generation, most evaluations focus on outcome-level metrics, such as final performance or aggregate success.

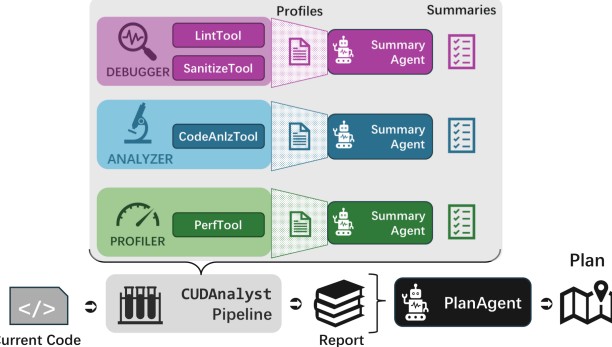

*Figure 2.* Overview of `CUDAnalyst`. Structured feedback reports serve as the sole input to planning, enabling controlled intervention and attribution of feedback-to-plan decisions at fixed generations.

These metrics offer limited insight into how feedback informs planning at each generation and *cannot disentangle the contributions or interactions of individual feedback signals*, while stochastic divergences and coupled trajectories further obscure causal effects.

## 3. `CUDAnalyst` Design and Evaluation

To analyze how feedback guides planning in self-evolving agents, we introduce `CUDAnalyst`, a unified *analysis layer* that decouples feedback from planning and enables controlled, generation-level interventions. By freezing program states and selectively manipulating feedback inputs, `CUDAnalyst` collects generation-level statistics and applies coalitional-style attribution to quantify both marginal contributions and interactions of feedback components. This design enables interpretable, intervention-based analysis of feedback-to-plan decisions during kernel evolution.

### 3.1. Feedback-to-Plan Analysis Layer

As shown in Fig. 2, `CUDAnalyst` separates feedback processing from planning while remaining agnostic to the surrounding self-evolving agent framework. Outputs from standard analysis tools are normalized into structured representations directly consumed by the planner. Reference program code is treated as a fixed prior and lies outside the attribution boundary; planning decisions are conditioned solely on feedback derived from the current program state.

The pipeline consists of three analysis modules: a *debugger*, *analyzer*, and *profiler*, each producing structured **profiles**. These profiles may be aggregated by a *SummaryAgent* into higher-level **summaries**, which together form a unified **report**. A dedicated *PlanAgent* consumes this report and outputs a high-level **plan**. This explicit separation exposes a clear evidence–decision boundary and enables controlled intervention on feedback inputs.

Both the SummaryAgent and PlanAgent are LLM-based and

operate under fixed prompts and decoding configurations across all generations and interventions. All components, including individual feedback sources, can be selectively enabled or disabled to support fine-grained attribution.

### 3.2. Generation-Level Feedback Intervention

End-to-end ablation is unreliable for self-evolving agents because perturbations introduced early in the evolutionary process propagate through iterative planning and confound attribution. To isolate the effect of feedback on planning at a fixed decision point, we adopt a **generation-level feedback intervention** protocol.

Specifically, we freeze the program state at selected generations, decoupling the current planning decision from the historical evolutionary trajectory (Ou et al., 2025; Chan et al., 2024; Desmond et al., 2025). References are an intrinsic part of the original program context and are frozen at each generation, with cached references reused across all feedback interventions. This ensures that all interventions share an identical program context and that attribution is not confounded by reference resampling.

At each frozen checkpoint, we perform controlled patching interventions (Zhang & Nanda, 2024) on the planning context by selectively withholding or injecting specific feedback components (Bush et al., 2025), while holding the planner, prompts, decoding configuration, and evaluation pipeline fixed. Differences in planning outcomes can therefore be attributed solely to changes in the feedback context.

This protocol avoids cross-generation interference and eliminates the need for full trajectory re-execution, enabling conditional attribution of feedback effects at fixed generations. Implementation details are provided in App. D.2, and a detailed analysis of the computational budget and sample efficiency is provided in App. D.3.

### 3.3. Evaluation Metrics

We report **generation-level statistics** computed immediately after each generation. For a fixed generation $g$, frozen program samples are evaluated using the same LLM, identical evaluation pipeline, and a fixed number of code-generation retries, held constant across samples and generations.

Execution-level outcomes follow the criteria of Ouyang et al. (2025):

- *compiled*: produces a runnable executable;
- *pass*: satisfies all functional validations;
- *fast*: outperforms a baseline implementation.

Each execution is assigned the highest satisfied criterion, with failures labeled *failed*. These criteria induce a partial ordering over executions. A sample is considered successful if at least one execution satisfies a criterion. Generation-level statistics are computed by aggregating sample-level indicators under the fixed execution budget, analogous to pass@$k$ (Chen et al., 2021) while preserving generation as the unit of attribution.

### 3.4. Component Attribution via Coalitional-Style Attribution

Inspired by interaction-based analyses of LLMs (Qin et al., 2026), which decompose outputs into interactions to reveal fine-grained sensitivities, we formalize feedback attribution as a cooperative game $\mathcal{G} = (N, v)$ (Shapley, 1952), where each player corresponds to a feedback context component. The characteristic function $v : 2^N \to \mathbb{R}$ maps a subset of components to the expected generation-level performance, estimated via repeated rollouts under fixed prompts and decoding configurations (Yang et al., 2025; Liu et al., 2024c). This formulation allows us to attribute planning outcomes to individual feedback components and their interactions.

We quantify marginal contributions using a coalitional attribution based on the Banzhaf value $\phi_i$ (Banzhaf III, 1964),

$$\phi_i(v) = \frac{1}{2^{|N|-1}} \sum_{S \subseteq N \setminus \{i\}} \big[ v(S \cup \{i\}) - v(S) \big], \quad (1)$$

By definition, $\phi_i(v)$ averages over all subsets (coalitions) of feedback components, ignoring ordering. In our setting, when forming a plan decision, all enabled feedback components appear simultaneously and are treated equally in the input, motivating the use of the Banzhaf value rather than the Shapley value. The baseline $v(\emptyset)$ corresponds to the coalition where none of the feedback components considered in the current research question are present.

To characterize non-additive dependencies, we compute the pairwise interaction term (Grabisch & Roubens, 1999)

$$\sigma_{ij} = v(\{i, j\}) - v(\{i\}) - v(\{j\}) + v(\emptyset), \quad (2)$$

where $\sigma_{ij} > 0$ indicates complementarity and $\sigma_{ij} < 0$ indicates redundancy or competition. Together, $\phi_i$ and $\sigma_{ij}$ provide fine-grained, intervention-based attribution of how individual feedback components and their interactions influence planning outcomes during kernel evolution.

## 4. Empirical Study

We conduct an empirical study to examine how feedback-conditioned planning affects generation-level decision outcomes in self-evolving LLM agents, while explicitly isolating these effects from trajectory-dependent drift or policy adaptation.

*Table 1.* Research Questions in Empirical Study

| RQ | Objective |
|---|---|
| **RQ0** (4.1) Planning | Validate when *explicit planning* is beneficial under different feedback conditions. |
| **RQ1** (4.2) Tool feedback | Measure the contribution of each tool's *feedback* to planning decisions. |
| **RQ2** (4.3) Tool summary | Assess if *summaries* improve decision quality over raw profiles. |
| **RQ3** (4.4) Distillation | Study whether *plans* produced by strong models can guide weaker ones. |

We adopt a **trajectory-freezing** evaluation protocol: at each generation, all program samples produced by the evolutionary process are frozen and independently re-evaluated under *controlled feedback configurations*, without re-running the evolutionary process. This protocol enables controlled interventional attribution of outcome differences to feedback interventions applied at fixed generations, rather than to learning or adaptation of the planning policy (App. B.1).

Under this protocol, we investigate four research questions summarized in Tab. 1: when explicit planning is beneficial (RQ0), how heterogeneous tool feedback contributes to planning outcomes (RQ1), whether summarization mediates feedback complexity (RQ2), and whether plans produced by strong models can guide weaker ones (RQ3).

Across all experiments, we report generation-level success rates aggregated over the entire PolyBench-ACC (Grauer-Gray et al., 2012) suite with 10 independent runs, providing a consistent basis for comparison across research questions; shaded regions in figures indicate 95% confidence intervals over these runs. Detailed breakdowns of fast and pass rate changes for each RQ are provided in the corresponding appendices.

### 4.1. RQ0: When Is Explicit Planning Beneficial?

We investigate whether explicit planning should be decoupled from code generation, and under what conditions such decoupling is beneficial. Our central hypothesis is that explicit planning is neither necessary nor beneficial per se, but becomes conditionally effective only when it encodes aligned external feedback, where it functions as a feedback-aligned abstraction that structures generation-level decisions under fixed feedback.

To test this hypothesis, we cross two factors: planning structure (implicit vs. explicit) and feedback availability (none vs. full). Implicit planning follows OpenEvolve's default full rewrite loop, in which planning and code generation are entangled within a single step. In contrast, CUDAnalyst introduces an explicit *PlanAgent* that maintains a persistent planning state across generations, enabling feedback-conditioned decision reuse.

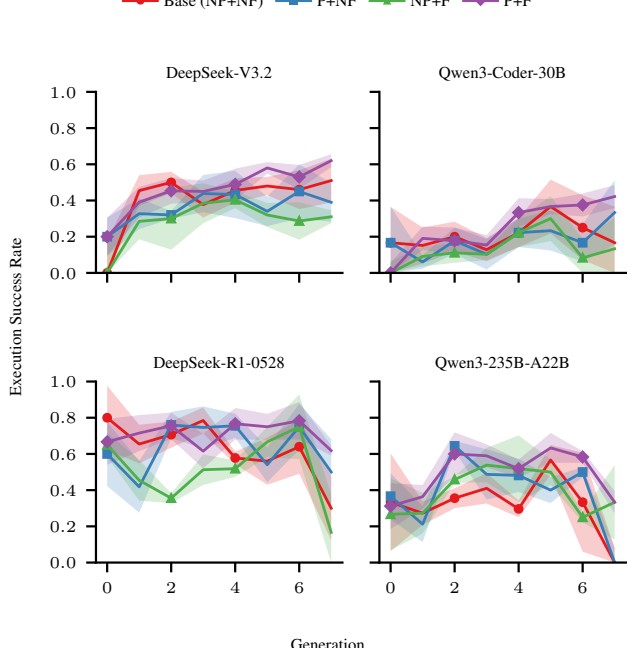

*Figure 3.* Per-generation execution success rate under different planning–feedback configurations. Explicit planning without feedback (P+NF) fails to improve execution outcomes, whereas feedback-grounded planning (P+F) yields stable gains across generations, particularly for weak-reasoning models.

Fig. 3 shows that explicit planning without feedback consistently degrades performance, while feedback-grounded planning improves execution success. This indicates that explicit planning primarily serves as an interface for organizing and reusing feedback at the outcome level, rather than as an independent enhancement of intrinsic reasoning capacity. Generation-level attribution results are reported in App. B.2.

To rule out confounds arising from increased token budget or superficial textual structure, we conduct two counterfactual controls. First, we replace planner outputs with a fixed, content-free template (DummyPlan, DP), removing semantic planning while preserving token length. Second, we randomize feedback assignments within each generation (P+RF), preserving feedback volume but destroying alignment. Experimental details are provided in the same appendix.

As shown in Fig. 4, the effectiveness of explicit planning depends jointly on feedback alignment and model reasoning capacity. Strong models are largely insensitive to planner semantics, while weaker models benefit from structured, feedback-aligned plans as an external decision scaffold under the evaluated protocol. Across all models, misaligned feedback degrades performance, confirming that explicit planning is beneficial only insofar as it encodes feedback-aligned decision structure rather than additional planning

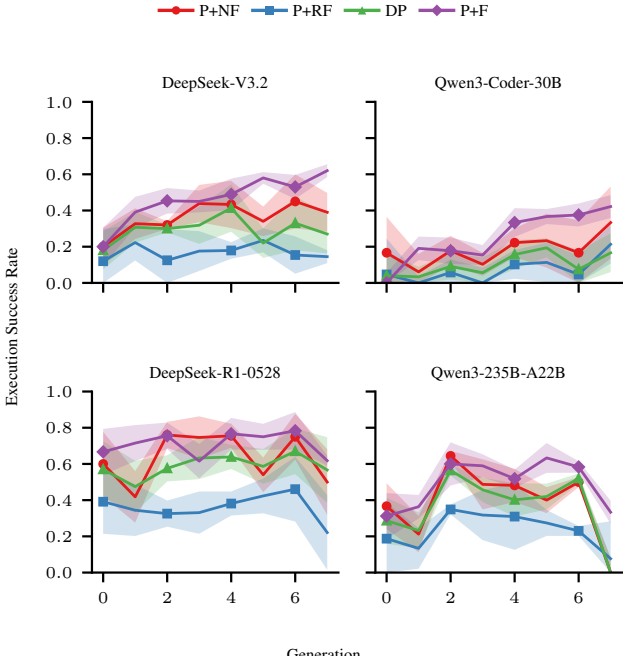

*Figure 4.* Counterfactual controls for RQ0. DummyPlan (DP) mainly degrades weak models, while randomized feedback (P+RF) consistently harms all models, highlighting the importance of aligned planning signals.

tokens.

These results indicate that explicit planning functions as a feedback-conditioned decision interface rather than an independent enhancement of intrinsic reasoning capability.

### 4.2. RQ1: How Does Tool Feedback Influence Planning Decisions?

Self-evolving agents typically rely on multiple feedback sources, yet it remains unclear whether planning decisions are driven primarily by a dominant tool or by the joint availability of multiple feedback components. This distinction is critical for understanding whether planning operates as a tool-specific heuristic or as an integrative decision process.

We perform generation-level **coalitional-style attribution analysis**, decomposing execution outcomes into marginal contributions and higher-order interaction effects (Fig. 5; App. B.3). Each feedback component is treated as a coalition member under fixed planning and evaluation conditions. These attributions reflect outcome-level effects under controlled feedback interventions at fixed generations, rather than global causal mechanisms of the planner or the evolutionary process.

Fig. 5 illustrates how feedback influences planning across generations. In early generations (0-2), marginal contributions from individual components are sparse and volatile, indicating unstable guidance under early-generation pro-

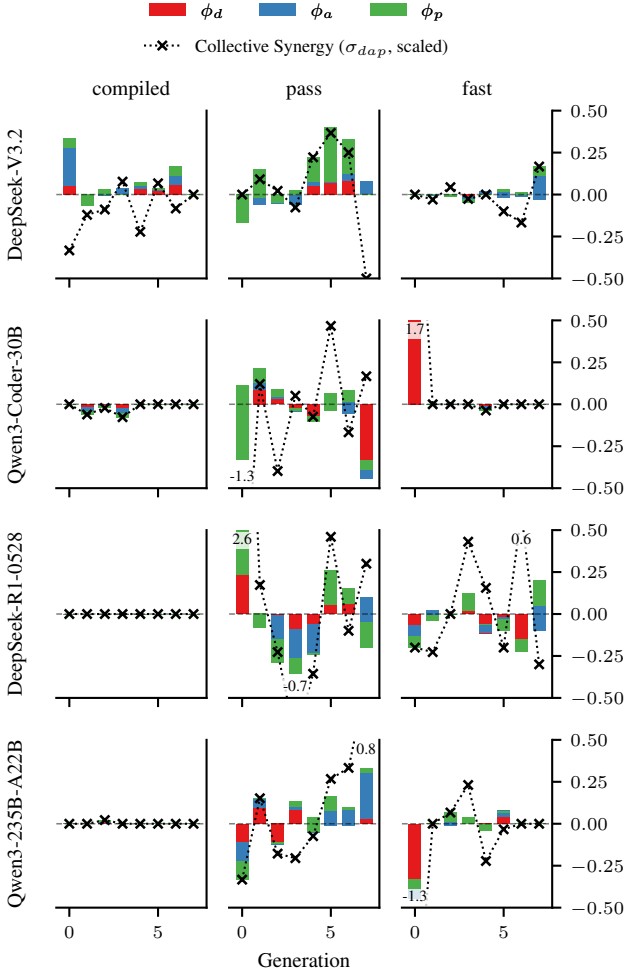

*Figure 5.* Per-generation coalitional-style decomposition of feedback components in planning decisions. Stacked bars show marginal contributions of the debugger ($\phi_d$), analyzer ($\phi_a$), and profiler ($\phi_p$); the dashed line denotes the higher-order interaction term $\sigma_{dap}$. Rows correspond to models and columns to execution metrics (compiled, pass, fast). Values are clipped for visualization.

gram states. In later generations (5-7), the contribution of interaction terms to execution outcomes increases over generations, reflecting the fact that later-generation programs expose more coupled failure modes that require joint feedback signals.

Distinct functional roles emerge across execution metrics: the analyzer consistently supports compilation, the profiler increasingly drives fast execution in later generations, and the debugger exhibits minimal marginal contribution but contributes through interactions. Interaction terms strengthen over generations, indicating that planning outcomes increasingly depend on the joint availability of multiple feedback components.

These attributions reflect outcome-level effects under controlled feedback interventions rather than internal causal mechanisms of the planner. Taken together, planning deci-

sions are not dominated by any single feedback source but instead reflect stable coordination among multiple feedback components.

## 4.3. RQ2: Do Summaries Improve or Replace Planning?

Building on RQ1, we examine whether alternative feedback representations, specifically, *summaries*, can improve planning quality or partially substitute for explicit planning. Prior work on self-evolving LLM agents often distills raw feedback into concise guidance to facilitate downstream decisions (Zaeed et al., 2025; Vatai et al., 2025). We ask whether such abstraction enhances planning or merely compresses feedback without capturing its full decision structure.

Summaries are generated deterministically from frozen program states and raw feedback using a fixed prompting template, introducing abstraction without new external signals or cross-generation state. They introduce inductive bias through abstraction but do not incorporate new external signals, and are therefore treated as a fixed feedback representation transform.

### RQ2.1: Does Summarized Feedback Improve Planning?
To isolate the effect of summarization, we hold the planning policy fixed and vary only the feedback representation. Fig. 6a shows that summarized feedback (P+S) consistently improves overall success for weaker models (DeepSeek-V3.2, Qwen3-Coder-30B), while gains for stronger models (DeepSeek-R1-0528, Qwen3-235B-A22B) are smaller and less consistent.

These findings indicate that summarization primarily benefits weaker models by reducing representational burden, whereas models with sufficient planning capacity derive limited additional gains. This aligns with the per-generation analysis in App. B.4, where summarized feedback notably accelerates *fast* successes.

### RQ2.2: Can Feedback Summarization Substitute for Explicit Planning?
We further compare agents using summarized feedback with implicit planning (NP+S) to those combining explicit planning with summaries (P+S). Across all models, explicit planning consistently outperforms summaries alone (Fig. 6b). While strong-reasoning models derive modest benefits from summaries, planning remains the dominant factor shaping execution success.

These results demonstrate that under the evaluated protocol, summarization compresses content, whereas explicit planning maintains and reuses decision structure across generations, a role that summaries alone cannot fulfill. Instead, summarization improves the accessibility of feedback representations, while explicit planning *remains necessary* to exploit decision structure under the evaluated protocol.

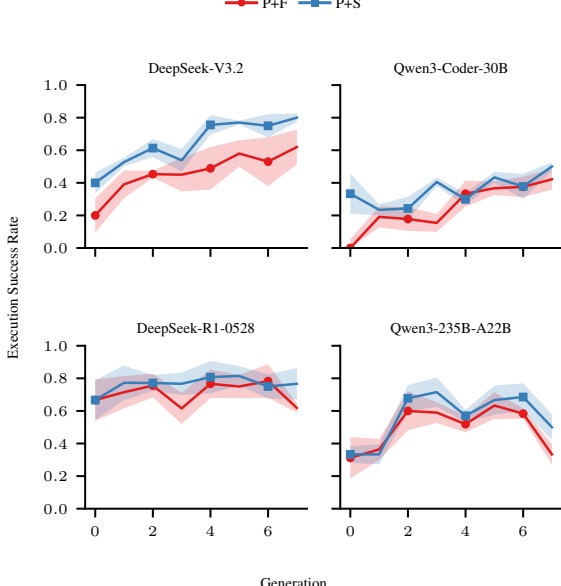

*(a)* P+F vs. P+S (effect of summarization under fixed planning).

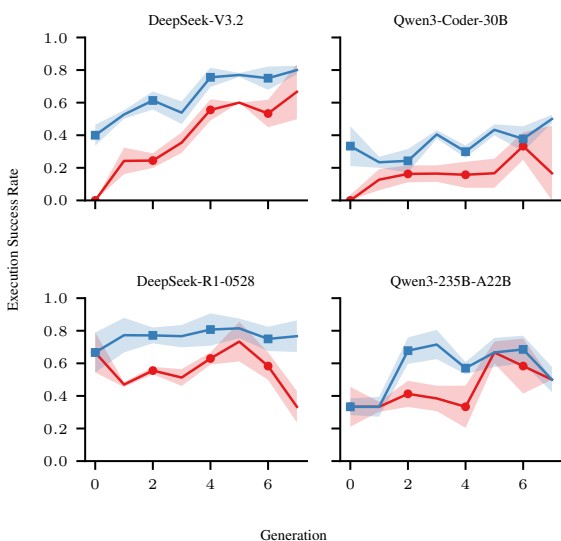

*(b)* NP+S vs. P+S (effect of planning under summarized feedback).

*Figure 6.* Decomposing the roles of summarization and planning in RQ2. Summarization improves feedback accessibility, while explicit planning remains necessary to exploit decision structure.

## 4.4. RQ3: Can Strong Reasoning Models Guide Weak Models via Explicit Plans?

To evaluate whether explicit plans function as **transferable decision abstractions**, we test if plans generated by strong models can effectively guide weaker models in code generation. Specifically, we inject plans from a strong model

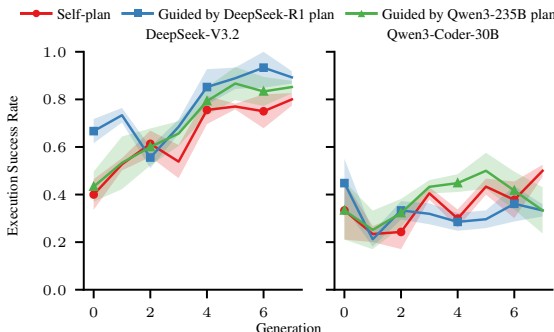

*Figure 7.* Effect of strong-to-weak plan distillation on code generation success. Injecting plans from strong reasoning models consistently improves weak models over their self-generated plans. Distillation within the same model family yields larger gains, suggesting improved compatibility between plan representations and downstream generation.

into a weaker model's context under identical task settings, while holding context length and decoding configurations fixed to isolate the effect of the plan's semantic content. By comparing execution success against baselines where the weak model generates its own plans, we assess the extent to which explicit planning acts as a model-agnostic decision interface rather than a mere reflection of internal reasoning capability.

As shown in Fig. 7, strong-to-weak plan injection consistently improves execution success for weak models, indicating that gains arise from *structured decision content* rather than extra context or computation. Occasional cases where weak models surpass their guides are largely due to improved *pass* rates: weak models follow structured plans more conservatively, while strong models may trade some *pass* for subsequent *fast* success (App. B.5).

Performance gains are larger when strong and weak models belong to the same model family (e.g., DeepSeek-R1-0528 → DeepSeek-V3.2, Qwen3-235B-A22B → Qwen3-Coder-30B). This pattern suggests that shared training distributions or representational structures improve plan interpretability, enabling more effective utilization of transferred decision abstractions.

### 4.5. Summary of Empirical Findings

Our empirical study yields four connected findings on the role of explicit planning in self-evolving agents:

- **Explicit planning is effective only when grounded in feedback:** Planning without feedback degrades performance, while feedback-aligned planning yields stable generation-level improvements, indicating that planning operates as a feedback-conditioned decision interface (RQ0).

- **Planning effectiveness arises from multi-tool inter-**

**actions:** Analysis dominates early compilation, profiling contributes to later performance gains, and debugging mediates interactions, with planning outcomes reflecting stable dependencies on joint feedback availability under controlled intervention (RQ1).

- **Summarization facilitates but does not replace planning:** Summaries disproportionately benefit weaker models by reducing feedback complexity, but explicit planning remains necessary for effective decision-making (RQ2).

- **Explicit plans function as partially transferable decision interfaces:** Plans generated by strong models consistently improve weaker models, especially within the same model family, demonstrating that explicit plans act as transferable decision representations across models (RQ3).

## 5. Generalization as Invariance of Feedback-Conditioned Planning Decisions

Building on Sec. 4, we test whether feedback-conditioned planning remains stable under controlled distribution shifts, using the weak-reasoning model DeepSeek-V3.2 as a unified testbed. Analyses focus on qualitative planning behaviors, attribution patterns, and structural dependencies rather than absolute performance. Experimental protocols are in App. C.1.

### 5.1. Consistency across Backbone Trajectories

To assess generalizability, we fix **DeepSeek-V3.2** as the evaluator while varying the sources of frozen **backbone trajectories**, ranging from weaker open-source models such as **Kimi-K2** and **MiniMax-M2.5** to the stronger proprietary **Gemini-2.5-Pro**. This setup isolates the effect of the underlying code environment on planning behavior. Across all settings, our central finding remains consistent: *tool synergy is largely architecture-invariant* (Fig. 8).

Importantly, the nature of synergy evolves with backbone capability. For weaker trajectories, synergistic effects are primarily associated with correctness-oriented reasoning, whereas for stronger trajectories (e.g., Gemini-2.5-Pro), synergy increasingly concentrates on performance-oriented optimization. This trend suggests that CUDAnalyst captures stable planning dynamics across heterogeneous agent backbones while remaining sensitive to shifts in optimization focus as model capability improves.

### 5.2. Robustness across Diverse Workloads

We evaluate whether the planning behaviors identified in Sec. 4 persist across diverse CUDA workloads: NPB-GPU (NPB) (Araujo et al., 2020), XSBench (Tramm et al., 2014),

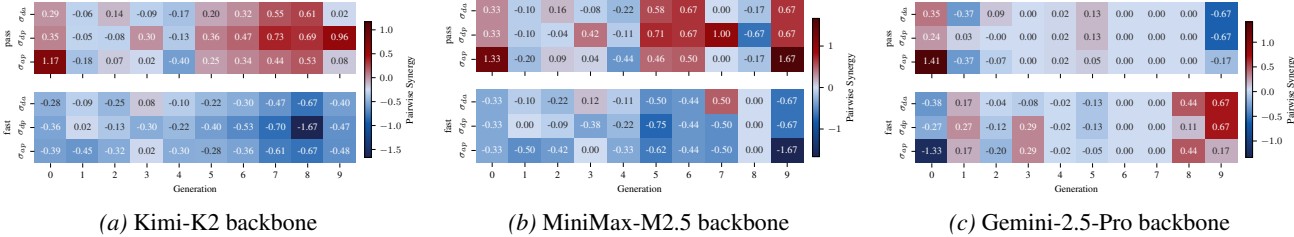

*(a)* Kimi-K2 backbone        *(b)* MiniMax-M2.5 backbone        *(c)* Gemini-2.5-Pro backbone

*Figure 8.* Pairwise tool synergies under different frozen backbone trajectories while using DeepSeek-V3.2 as the evaluator.

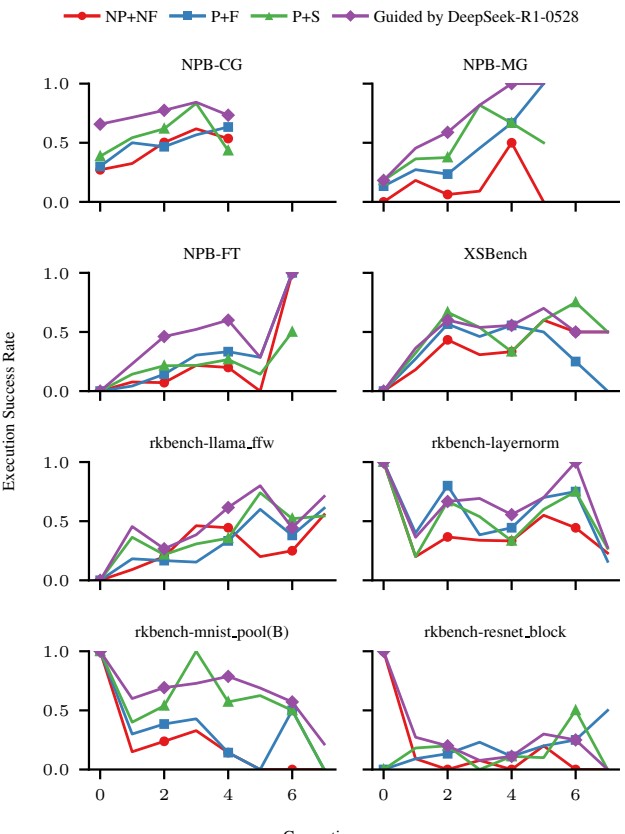

*Figure 9.* Generation-level execution success rates across diverse CUDA workloads for DeepSeek-V3.2.

and robust-kbench (rkbench) (Lange et al., 2025) (Table 9; details in App. C.2). Analyses focus on the stability of qualitative roles and interactions of feedback components rather than absolute performance.

Across workloads, we observe consistent qualitative patterns (Fig. 9):

- Explicit planning improves outcomes only when grounded in informative feedback.

- Performance gains arise from structured interactions among multiple feedback signals: analysis benefits early compilation, profiling drives later gains, and de-

bugging mediates dependencies.

- Summarization accelerates early progress, especially for weaker models, but cannot replace explicit feedback in later generations.

While the magnitude of improvements varies: NPB converges faster, XSBench is sensitive to noisy feedback, and rkbench highlights recovery under heterogeneity, the qualitative patterns of feedback-conditioned planning remain stable. Detailed breakdowns are shown in App. C.3.

### 5.3. Invariance to Reference Induction Regimes

We examine whether feedback-conditioned planning is sensitive to the reference induction mechanism. To do so, we evaluate DeepSeek-V3.2 on the XSBench workload, generating distinct reference distributions using varied evolutionary operators, including EoH (Liu et al., 2024a), MCTS-AHD (Zheng et al., 2025), LHNS (Xie et al., 2025), and hill-climbing, while keeping the workload constant.

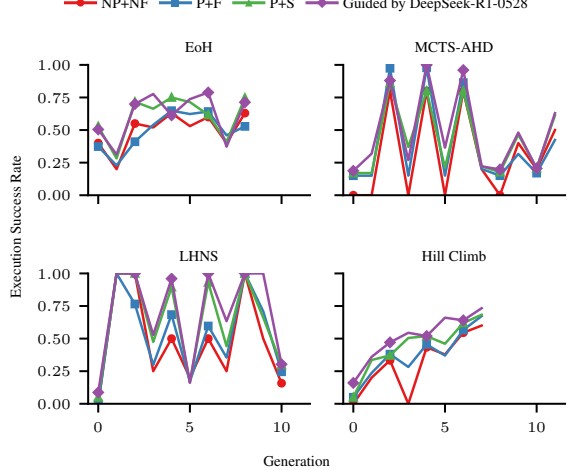

*Figure 10.* Execution success rates for DeepSeek-V3.2 across diverse reference induction regimes. The synchronized trajectories reveal a consistent model affinity for summarized feedback ($P+S$), which facilitates rapid planning convergence toward high-level heuristics rather than stochastic exploration.

As shown in Fig. 10, execution dynamics exhibit strong structural invariance across all regimes, with trajectories

remaining largely deterministic and synchronized despite differing evolutionary operators. Summarized feedback consistently stabilizes performance and outperforms raw feedback, even under greedy hill-climbing, suggesting that the model inherently favors high-level heuristic refinement and fast planning convergence (App. C.4), effectively decoupling planning logic from the underlying evolutionary dynamics.

### 5.4. Cross-Domain Study: CPU-based Numba Optimization

To test whether our findings generalize beyond CUDA kernel synthesis, we extend the framework to a **CPU-based Numba N-body simulation**, shifting from GPU-oriented CUDA C++ generation to JIT-compiled Python optimization. In this setting, CUDA-specific diagnostics such as NCU profiling and Tree-sitter analysis are replaced with Python-native profiling tools including `cProfile` and `line_profiler`, while preserving the same planning-feedback interaction pipeline. This allows us to evaluate whether the observed planning behaviors are tied to CUDA-specific heuristics or reflect more general optimization principles.

As shown in Fig. 11, guided planning consistently shifts the optimization trajectory toward higher-performance regions focusing on classical CPU optimization such as SIMD-aware loop restructuring and improved cache locality, ultimately reaching a $12.5\times$ peak speedup for DeepSeek-R1 guidance. These results suggest that our strategic planning mechanism is largely tool-agnostic and generalizes across heterogeneous HPC optimization domains beyond the CUDA ecosystem.

### 5.5. From Invariant Insights to Actionable Design

To operationalize invariant planning patterns, we introduce `CuGEdit`, a modular plugin that can be integrated into self-evolving LLM agent frameworks. It leverages kernel-similarity-aware activation and feedback summarization at key stages. It further distills plans from stronger to weaker models to guide weaker agents and reduce token usage, thereby guiding the evolutionary search toward promising regions. Empirical validation via KernelBench Level 3 shows that `CuGEdit`-enhanced OpenEvolve achieves $2.08\times$ to $10.32\times$ speedup over `torch.compile`, surpassing both baseline and existing SOTA approaches (App. E).

## 6. Conclusion and Limitations

We study planning decisions in self-evolving LLM agents for CUDA kernel generation through a feedback-centric perspective, introducing `CUDAnalyst` to disentangle feedback from planning at the generation level. Through

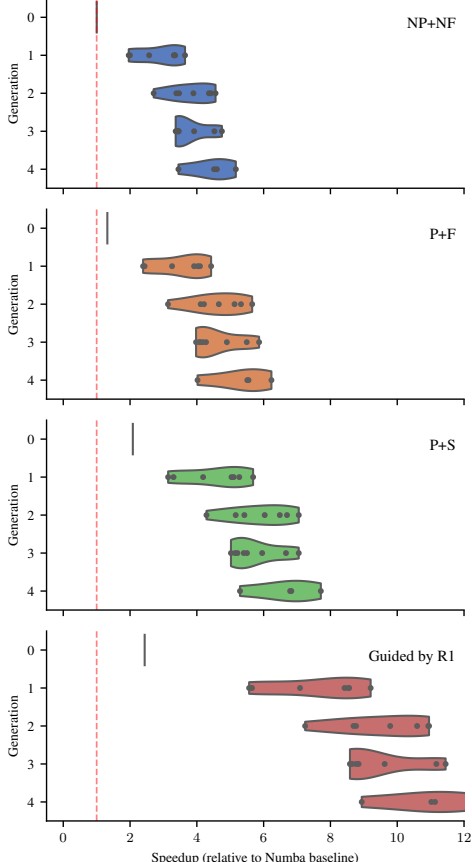

*Figure 11.* Generalization results on Numba N-body. Distributions of instantaneous speedup per generation. Red dashed lines indicate the $1.0\times$ baseline.

generation-level interventions and coalition analysis, we show that effective planning depends critically on grounded feedback, that tool effects interact compositionally, and that planning behavior exhibits structured, non-uniform dynamics. These trends remain consistent across workloads, backbone trajectories, evolutionary operators, and reference induction regimes, suggesting a robust and architecture-invariant feedback-to-plan structure largely decoupled from specific experience distributions. Together, these properties position `CUDAnalyst` as a fine-grained and computationally efficient framework for credit assignment in self-evolving LLM agents with coupled components such as Chen et al. (2026).

Our methodology freezes implicit memory states to isolate the immediate causal effect of feedback on planning decisions. This abstraction is necessary as attributing optimization gains to the semantic evolution of CUDA kernel state elements remains an open compiler research challenge (Deiana et al., 2023; Ivanov et al., 2024), making rigorous cross-generation attribution difficult. We therefore focus specifically on feedback-to-plan decisions in self-evolving LLM agents under controlled frozen-trajectory settings.

## Acknowledgements

This work was supported by the Brain Science and Brain-like Intelligence Technology — National Science and Technology Major Project (Grant No. 2025ZD0215500), the National Key Research and Development Program of China (Grant No. 2025YFB3003200), the Jiangsu Provincial Science and Technology Program (Grant No. BE2023005-3), and the Tsinghua University Initiative Scientific Research Program (Grant No. 2022Z11ZRB002). We gratefully acknowledge Huawei for supporting this work with the Ascend 910 series computing infrastructure. We also thank Jianmin Wu and Annan Li from Baidu for insightful discussions in shaping this work.

## Impact Statement

This paper presents a systematic framework for analyzing self-evolving LLM agents in high-performance computing (HPC) tasks. Our approach provides a principled methodology for quantifying the causal effects of heterogeneous feedback on agentic planning decisions, offering new insights into the internal decision-making dynamics of LLMs in complex, long-horizon code optimization tasks.

By disentangling feedback attribution from trajectory drift through the proposed intervention protocol, this work establishes a more stable and computationally efficient evaluation framework for studying the capabilities of language models in HPC kernel optimization and their use of tool synergies. Furthermore, our findings on cross-model plan transferability provide theoretical insights into collaborative optimization among heterogeneous LLM agents, with potential implications for the design of modular, interpretable, and scalable automated software engineering systems.

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

## A. Existing Work

Existing self-evolving agents for high-performance kernel generation primarily rely on two forms of direct feedback: runtime profiling (Zaeed et al., 2025; Zhang et al., 2026b) and static code analysis (Tschand et al., 2025; Merouani et al., 2025; Vatai et al., 2025). Runtime profiling tools (e.g., Nsight Compute) expose execution-level metrics such as memory behavior and occupancy, while static analyzers built on domain-specific compilers (Verdoolaege & Grosser, 2012; Baghdadi et al., 2019) extract loop structure, affine accesses, and dependencies to guide optimization.

Some systems additionally incorporate historical kernels from the evolution trajectory as contextual guidance. These kernels are typically selected using performance-diversity balancing (Novikov et al., 2025), lineage-based tracing (Dong et al., 2026), similarity-aware pruning (Guo et al., 2026), or strategy-aware retrieval (Kong et al., 2026), and function as long-horizon search priors rather than generation-local optimization evidence. Consequently, these mechanisms operate at a different temporal and causal granularity than the generation-local feedback-to-plan decisions studied in this work.

To manage heterogeneous and low-level feedback, several approaches adopt multi-agent designs (Wei et al., 2025; Lei et al., 2025; Nagaitsev et al., 2025; Chen et al., 2025), where auxiliary agents distill profiling outputs and contextual signals into higher-level guidance for code generation (Xiao et al., 2025). While this decomposition stabilizes iterative refinement, the causal contributions of individual feedback components to planning decisions remain implicit.

Evaluation in prior work commonly relies on end-to-end ablation, where the evolution is restarted or resumed under different feedback configurations, and the best-achieved kernels are reported (Novikov et al., 2025; Zhang et al., 2026a; Liu et al., 2024b; Lange et al., 2026; Assumpção et al., 2026). Such trajectory-level analyses obscure when specific feedback signals take effect and how their interactions shape generation-level planning behavior.

## B. Supplementary Empirical Study Details

### B.1. Empirical Study Protocol

**Tasks and Workloads** We use PolyBench-ACC[1] as a controlled workload suite in which feedback signals can be consistently reproduced across runs, enabling feedback-level intervention and attribution at fixed generations. PolyBench-ACC consists of fundamental computational kernels designed to expose the effects of compiler optimizations on loop nests, memory accesses, and dependent computations

(Grauer-Gray et al., 2012). This provides a reproducible testbed where feedback signals can be reliably isolated and analyzed.

For the empirical study, the full PolyBench-ACC suite is used, with MINI_DATASET for correctness checking and SMALL_ through EXTRALARGE_DATASET for performance evaluation. For each dataset size, we perform three warm-up runs followed by ten measured runs, reporting relative speedup with respect to the official PolyBench-ACC implementation. Overall improvement is defined as the minimum speedup across all dataset sizes.

**Evolutionary Configuration and Trajectory Freezing** We conduct the experiments using OpenEvolve [2] (Sharma, 2025), an open-source implementation of the AlphaEvolve self-evolving agent (Novikov et al., 2025), on an NVIDIA RTX4090 GPU. OpenEvolve implements an evolutionary process in which a population of programs is iteratively rewritten by a planner conditioned on feedback, evaluated, and selected across generations. Evolution may proceed over multiple islands, where each island maintains an independent population, and optional migration introduces additional stochasticity through cross-island selection.

We adopt the official configuration for the MLX Metal kernel optimization task (Tab. 2) and the system prompt shown in Prompt B.1 throughout the study. All runs strictly reuse OpenEvolve's default full-rewrite-method template, with no modifications to its structure or instructions; feedback signals and plan decisions are parsed and recorded as artifacts.

All empirical studies are conducted under a **single-island** evolutionary configuration with an extended iteration budget (200 iterations). This induces a linear parent–child trajectory, eliminating population-level stochasticity introduced by migration or parallel evolution, and allows planning behavior to be analyzed as a temporally ordered sequence conditioned solely on feedback signals.

Under this configuration, we record the complete evolutionary trajectory and group frozen program samples by generation. These frozen generations serve as fixed starting points for controlled re-evaluation, allowing feedback signals to be selectively injected or removed without re-running evolution.

**Generation-level Feedback Intervention** To isolate the effect of feedback on planning, we re-evaluate all program samples from each frozen generation under controlled feedback configurations. Feedback signals are selectively injected, removed, or summarized at the planner input, while execution, compilation, and evaluation procedures remain

---

[1] https://github.com/cavazos-lab/PolyBench-ACC

[2] https://github.com/algorithmicsuperintelligence/openevolve

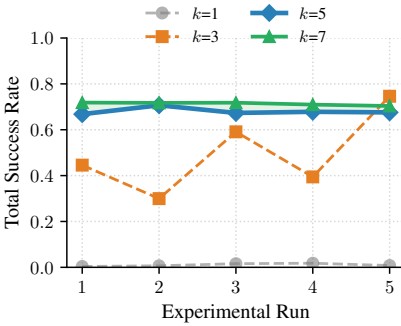

*Figure 12.* Total success rate for each experimental run on the `3DCONV` kernel after replaying frozen program samples with DeepSeek-V3.2. Lines show different $k$ settings, and the shaded area highlights the gap between $k = 5$ and $k = 7$. Each point represents the pass rate over all samples, showing that increasing $k$ beyond 5 provides only marginal gains.

*Table 2.* OpenEvolve config for MLX Metal kernel optimization.

| Parameter | Value |
|---|---|
| Checkpoint Interval | 10 |
| Population Size | 25 |
| Archive Size | 12 |
| Elite Selection Ratio | 0.3 |
| Explore/Exploit Ratio | 0.35/0.65 |

unchanged. Interventions at fixed generations prevent feedback effects from propagating through subsequent evolutionary steps, allowing direct attribution of planning decisions to specific feedback components.

Program samples are evaluated with up to five code-generation attempts ($k = 5$), counting success if any attempt achieves *pass* or *fast* (pass@5, fast@5). This choice balances stochastic coverage with evaluation cost: as shown in Fig. 12, increasing $k$ beyond 5 provides only marginal gains. The trend shown is representative of typical kernels in PolyBench-ACC.

**Models and Trajectory Decoupling**  We evaluate four LLMs spanning a range of reasoning capacity: DeepSeek-V3.2 with the thinking mode disabled and Qwen3-Coder-30B-A3B as weaker models, and DeepSeek-R1-0528 and Qwen3-235B-A22B-32K as stronger models. To avoid confounding attribution with a model's own coding style, all feedback interventions are applied to a fixed backbone trajectory $\mathcal{R}^*$ generated by a third-party model (GLM-4.5-Air), selected to provide sufficient optimization and debugging headroom for feedback effects to manifest. Trajectories that are already near-optimal and exhibit a ceiling effect, where feedback induces negligible planning changes, are excluded, as they do not admit meaningful attribution.

Output only the final code, unless an explicit explanation is requested.

# Optimization Rules

1. Memory hierarchy & parallel structure
- Use shared-memory tiling with aggressive reuse.
- Use `cp.async` for global to shared transfers and apply double-buffered pipelines.
- Fuse elementwise ops to eliminate redundant global traffic.
- Avoid shared-memory bank conflicts; apply padding/skew when required.
- Use vectorized loads/stores (`float4`, `int4`) when alignment allows.

2. Tensor Core compute
- Use WMMA or `mma.sync` paths with hardware-aligned MMA tiles (e.g., 16×16×16).
- Expose multiple independent MMA ops per warp to increase ILP (instruction-level parallelism).
- Use FP32 accumulation for mixed-precision math.

3. Loop transformations
- Apply in this order: Tile, Unroll, Skew/Permute, Double-buffer.
- Pick tile sizes that fit SMEM and register budgets (e.g., 64×64×16 as baseline).
- Fully unroll the inner K-loop to deepen ILP.

4. Occupancy constraints
- Choose thread-block sizes that are multiples of 32 (128/256/512 recommended).

5. Micro-optimizations
- Use inline PTX only where profiling would show unavoidable hotspots.
- Provide a portable CUDA path and an architecture-tuned fast path.

### B.2. Attributing the Benefits of Explicit Planning to Feedback (RQ0)

**Counterfactual Controls for Feedback-Aligned Planning** To disentangle feedback-aligned planning from superficial prompt or budget effects, we introduce two counterfactual controls that selectively remove semantic alignment while preserving surface structure. For *DummyPlan* (DP), planner outputs are replaced with a fixed, content-free template (Prompt B.2) that matches the original plan in length and format but carries no task-specific information. This controls for token budget, prompt structure, and planner invocation.

To test whether gains arise from feedback availability rather than alignment, we introduce *Randomized Feedback* (P+RF), where feedback reports are randomly reassigned across programs within the same generation. This preserves feedback volume and distribution while breaking correspondence to the program state.

Together, these controls rule out explanations based on additional tokens or feedback quantity, showing that explicit planning is beneficial only when feedback is semantically aligned with the current program context.

**Quantifying via Banzhaf Values**  We quantify the contributions of explicit planning ($P$) and external feedback ($F$) at the level of generation. For each generation $g$ and context $S \subseteq \{P, F\}$, let $v_g(S)$ denote the fraction of samples in generation $g$ that achieve at least one successful execution (*pass* or *fast*) under a fixed execution budget.

This yields four payoffs, $v_g(\emptyset)$, $v_g(P)$, $v_g(F)$, and $v_g(PF)$, forming a two-player cooperative game. For a two-player game, the Banzhaf values admit closed-form expressions that average marginal contributions over all coalitions with equal weight.

For generation $g$, the contributions of planning and feedback are

$$\phi_F^{(g)} = \frac{1}{2}\Big[(v_g(F) - v_g(\emptyset)) + (v_g(PF) - v_g(P))\Big], \quad (3)$$

$$\phi_P^{(g)} = \frac{1}{2}\Big[(v_g(P) - v_g(\emptyset)) + (v_g(PF) - v_g(F))\Big]. \quad (4)$$

These represent the average marginal contribution of each component across all inclusion orders. To quantify interaction effects, we define the synergy term:

$$\sigma_{FP}^{(g)} = v_g(FP) - v_g(P) - v_g(F) + v_g(\emptyset). \quad (5)$$

Across generations, planning accounts for the dominant marginal contribution, while feedback provides complementary gains. The consistently positive synergy terms indicate strong non-linear interactions, showing that the joint effect of planning and feedback often exceeds the sum of their individual contributions. This confirms that planning is most effective when grounded in aligned feedback, and that attribution methods ignoring interaction effects would substantially underestimate their joint benefits.

**Pass vs. Fast Breakdown of Explicit Planning Outcomes**
Fig. 13 presents a decomposition of generation-level outcomes under feedback-grounded explicit planning. While improvements are modest, *pass* shows the largest gains, with *fast* improvements being smaller and dependent on the models' intrinsic reasoning capabilities.

*Table 3.* Average generation-level Banzhaf values for feedback, planning, and their interaction.

*(a) DeepSeek-V3.2*

| | $\phi_F$ | $\phi_P$ | $\sigma_{FP}$ |
|---|---|---|---|
| 0 | 0.0 | 0.200 | 0.0 |
| 1 | -0.053 | -0.011 | 0.233 |
| 2 | -0.033 | -0.013 | 0.333 |
| 3 | 0.010 | 0.063 | 0.004 |
| 4 | 0.002 | 0.031 | 0.107 |
| 5 | 0.040 | 0.060 | 0.400 |
| 6 | -0.047 | 0.117 | 0.253 |
| 7 | 0.015 | 0.095 | 0.430 |

*(b) Qwen3-Coder-30B*

| | $\phi_F$ | $\phi_P$ | $\sigma_{FP}$ |
|---|---|---|---|
| 0 | -0.317 | -0.150 | 0.300 |
| 1 | 0.035 | 0.005 | 0.191 |
| 2 | -0.044 | 0.022 | 0.089 |
| 3 | 0.013 | 0.013 | 0.077 |
| 4 | 0.056 | 0.056 | 0.111 |
| 5 | 0.083 | 0.017 | 0.300 |
| 6 | 0.021 | 0.104 | 0.375 |
| 7 | 0.028 | 0.228 | 0.122 |

*(c) DeepSeek-R1-0528*

| | $\phi_F$ | $\phi_P$ | $\sigma_{FP}$ |
|---|---|---|---|
| 0 | 0.008 | -0.142 | 0.117 |
| 1 | 0.048 | 0.012 | 0.497 |
| 2 | -0.178 | 0.227 | 0.347 |
| 3 | -0.201 | 0.032 | 0.141 |
| 4 | -0.024 | 0.213 | 0.070 |
| 5 | 0.158 | 0.032 | 0.103 |
| 6 | 0.072 | 0.072 | -0.077 |
| 7 | -0.008 | 0.326 | 0.252 |

*(d) Qwen3-235B-A22B*

| | $\phi_F$ | $\phi_P$ | $\sigma_{FP}$ |
|---|---|---|---|
| 0 | -0.061 | 0.039 | 0.012 |
| 1 | 0.076 | 0.015 | 0.152 |
| 2 | 0.029 | 0.215 | -0.147 |
| 3 | 0.115 | 0.064 | -0.026 |
| 4 | 0.130 | 0.093 | -0.185 |
| 5 | 0.083 | -0.017 | 0.300 |
| 6 | 0.0 | 0.250 | 0.167 |
| 7 | 0.333 | 0.0 | 0.0 |

A consistent pattern is observed in the counterfactual control experiments (Fig. 14), showing that explicit planning benefits stem from feedback alignment. Notably, both P+RF and DP primarily affect *pass* outcomes, with minimal impact on *fast*.

**Prompt B.2: Dummy Plan**

### Executive Summary The system has completed the current execution cycle and performed a standard architectural review. The following summary provides a high-level overview of the diagnostic process and the general procedural framework applied to the codebase to maintain structural integrity and baseline operational standards.
### Critical Findings #### 1. **BLOCKER** - **Routine Verification**: All primary execution paths have been checked against the standard validation suite. No immediate catastrophic failures were flagged during the generic pass, though continuous monitoring is recommended as per standard protocol.
#### 2. **BOTTLENECK** *(General performance observations for standard optimization)*
- **Systemic Resource Utilization** - **Status**: Analysis of resource allocation suggests that throughput is subject to the theoretical limits of the current hardware configuration and environment settings.
- **Observation**: Standard performance metrics indicate that processing time is distributed across various computational modules without a singular anomalous outlier.

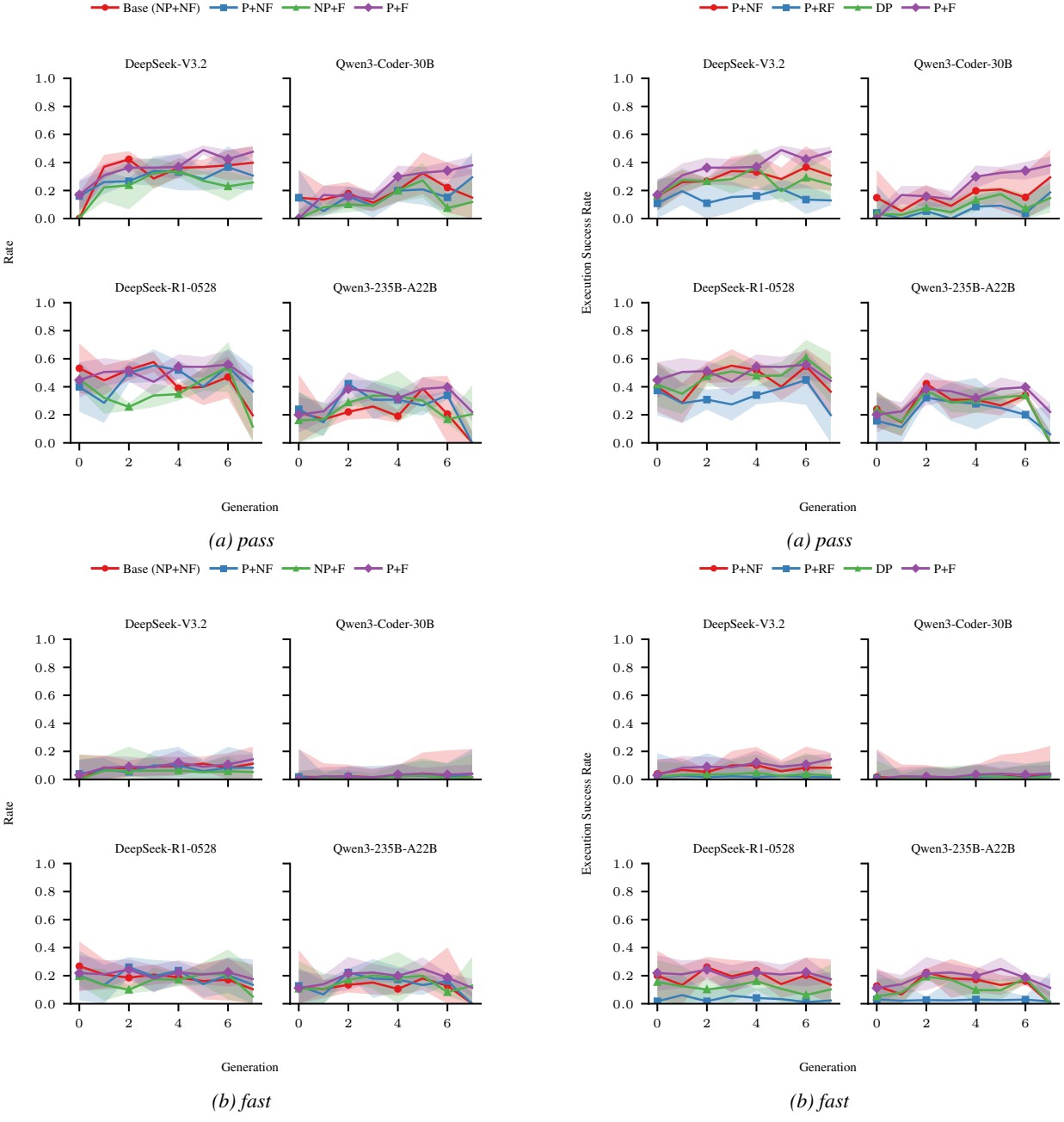

*Figure 13.* Generation-level breakdown of RQ0 outcomes. Weak models (top row) which mainly improve *pass*, while strong models (bottom row) improve both *pass* and *fast*.

*Figure 14.* Generation-level breakdown of RQ0 counterfactual outcomes. Without feedback, P+RF and DP

- **General Impact**: Maintenance of current through-put levels is expected unless structural modifications are implemented in future iterations.
- **Algorithmic Complexity Pass** - Routine complexity analysis confirms that the implemented logic follows the predefined control flow graph.
#### 3. **TECHNICAL DEBT** *(Standard maintenance considerations)*

- **Code Consistency**: Periodic refactoring is encouraged to ensure that all modules adhere to the latest documentation and style guidelines to prevent long-term degradation.

—

### Optimization Roadmap
#### [Step 1: Baseline Procedures (General Maintenance)]
1. **Standard Module Review** - **Action**: Con-

duct a systematic review of all active functions and data structures within the current scope. - **Rationale**: Ensures that the codebase remains aligned with general programming best practices and modularity standards.
2. **Data Flow Verification** - **Action**: Trace the movement of information across the primary interfaces to confirm consistent state transitions. - **Rationale**: Validates that the input-output relationship remains within expected nominal ranges.
#### [Step 2: Stability Enhancements]
3. **General Logic Refinement** - Apply standard optimization passes to the main execution loops to ensure no redundant operations are performed.
4. **Interface Synchronization** - Review inter-module communication protocols to ensure optimal handshake timing and resource locking.
#### [Step 3: Future Considerations]
5. **System Profiling** - Utilize standard profiling tools to gather telemetry on execution patterns for future comparative analysis.
6. **Documentation Update** - Ensure all recent changes are reflected in the technical specifications to maintain transparency for subsequent cycles.

### B.3. Quantifying Tool Contributions via Banzhaf Attribution and Synergy (RQ1)

To assess the individual and joint impact of the three tool modules in `CUDAnalyst`, namely the debugger ($d$), analyzer ($a$), and profiler ($p$), we analyze their contributions at the level of generation. For generation $g$ and any subset of tools $S \subseteq \{d, a, p\}$, let $v_g(S)$ denote the proportion of programs in generation $g$ that satisfy a given metric (*compile*, *pass*, or *fast*) under identical evaluation conditions.

This yields eight payoffs per generation: $v_g(\emptyset)$, $v_g(d)$, $v_g(a)$, $v_g(p)$, $v_g(da)$, $v_g(dp)$, $v_g(ap)$, and $v_g(dap)$, forming a three-player cooperative game.

The Banzhaf value of tool $t \in \{d, a, p\}$ at generation $g$ captures its average marginal contribution over all coalitions not containing $t$:

$$\phi_t^{(g)} = \frac{1}{|\mathcal{S}_t|} \sum_{S \in \mathcal{S}_t} \left[ v_g(S \cup \{t\}) - v_g(S) \right], \quad (6)$$

where $\mathcal{S}_t = \{S \subseteq \{d, a, p\} \setminus \{t\}\}$. Since the PlanAgent processes all tools' feedback simultaneously and no ordering over tools is assumed, we adopt a Banzhaf-style attribution, averaging marginal contributions uniformly over all coalitions rather than over player permutations.

To quantify interactions, we define the pairwise synergy between tools $t_1$ and $t_2$ as

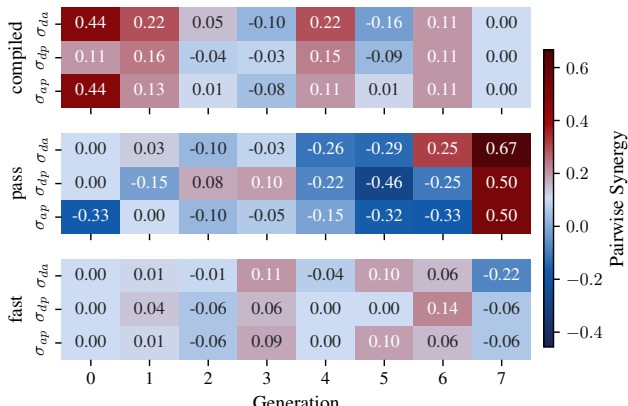

*Figure 15.* Pairwise synergies for DeepSeek-V3.2

$$\sigma_{t_1 t_2}^{(g)} = v_g(\{t_1, t_2\}) - v_g(t_1) - v_g(t_2) + v_g(\emptyset)$$
$$- \frac{1}{3} \sigma_{dap}^{(g)}, \quad (7)$$

and the three-way synergy among all tools is

$$\sigma_{dap}^{(g)} = v_g(dap) - \left( v_g(da) + v_g(dp) + v_g(ap) \right)$$
$$+ v_g(d) + v_g(a) + v_g(p) - v_g(\emptyset). \quad (8)$$

**DeepSeek-V3.2 (Fig. 15, Tab. 4)** The Banzhaf analysis reveals coherent and progressively stabilized tool utilization across generations. In early generations (0–2), *compiled* is driven by strong pairwise synergies, most notably between the debugger and analyzer ($\sigma_{da}$), indicating that combining static analysis with runtime debugging is particularly effective during initial exploration. As evolution progresses (3–5), individual Banzhaf values become more stable and positive, while pairwise synergies moderate, suggesting a transition from interaction-dominated gains to more reliable marginal contributions. In later generations (6–7), both Banzhaf values and synergies converge toward smaller magnitudes, indicating stabilization of the optimization process. A similar trend holds for *pass*, where analyzer-profiler interactions ($\sigma_{ap}$) dominate early improvements before giving way to more balanced contributions. In contrast, *fast* remains modest but persistent throughout. Overall, DeepSeek-V3.2 demonstrates sustained and compositional exploitation of multi-tool feedback, maintaining structured gains over longer evolutionary horizons rather than collapsing after early interaction effects.

**Qwen3-Coder-30B (Fig. 16, Tab. 5)** The Banzhaf analysis indicates substantially weaker and less compositional tool utilization than DeepSeek-V3.2, with a clear early–late generational split. In early generations (0–3), *compiled*

*Table 4.* Generation-level Banzhaf values for tool contributions and their interactions with DeepSeek-V3.2

*(a) compiled*

|  | 0 | 1 | 2 | 3 | 4 | 5 | 6 | 7 |
|---|---|---|---|---|---|---|---|---|
| $\phi_d$ | 0.056 | -0.010 | -0.007 | 0.038 | 0.037 | 0.022 | 0.056 | 0.0 |
| $\phi_a$ | 0.222 | -0.056 | 0.037 | -0.038 | 0.019 | 0.006 | 0.056 | 0.0 |
| $\phi_p$ | 0.056 | 0.066 | -0.030 | 0.0 | 0.019 | 0.006 | 0.056 | 0.0 |
| $\sigma_{da}$ | 0.444 | 0.222 | 0.052 | -0.103 | 0.222 | -0.156 | 0.111 | 0.0 |
| $\sigma_{dp}$ | 0.111 | 0.162 | -0.037 | -0.026 | 0.148 | -0.089 | 0.111 | 0.0 |
| $\sigma_{ap}$ | 0.444 | 0.131 | 0.007 | -0.077 | 0.111 | 0.011 | 0.111 | 0.0 |
| $\sigma_{dap}$ | -0.333 | -0.121 | -0.089 | 0.077 | -0.222 | 0.067 | -0.083 | 0.0 |

*(b) pass*

|  | 0 | 1 | 2 | 3 | 4 | 5 | 6 | 7 |
|---|---|---|---|---|---|---|---|---|
| $\phi_d$ | 0.0 | -0.061 | -0.037 | -0.064 | 0.056 | 0.072 | 0.083 | 0.083 |
| $\phi_a$ | -0.167 | 0.045 | -0.015 | 0.064 | 0.019 | 0.006 | 0.042 | -0.083 |
| $\phi_p$ | 0.167 | 0.167 | 0.052 | 0.026 | 0.148 | 0.322 | 0.208 | -0.0 |
| $\sigma_{da}$ | 0.0 | 0.030 | -0.096 | -0.026 | -0.259 | -0.289 | 0.250 | 0.667 |
| $\sigma_{dp}$ | 0.0 | -0.152 | 0.081 | 0.103 | -0.222 | -0.456 | -0.250 | 0.500 |
| $\sigma_{ap}$ | -0.333 | 0.0 | -0.096 | -0.051 | -0.148 | -0.322 | -0.333 | 0.500 |
| $\sigma_{dap}$ | 0.0 | 0.091 | 0.022 | -0.077 | 0.222 | 0.367 | 0.250 | -0.500 |

*(c) fast*

|  | 0 | 1 | 2 | 3 | 4 | 5 | 6 | 7 |
|---|---|---|---|---|---|---|---|---|
| $\phi_d$ | 0.0 | 0.005 | -0.007 | -0.009 | 0.019 | -0.017 | -0.014 | -0.028 |
| $\phi_a$ | 0.0 | -0.010 | -0.007 | -0.021 | -0.019 | 0.033 | 0.028 | 0.139 |
| $\phi_p$ | 0.0 | 0.005 | 0.015 | -0.021 | 0.0 | 0.017 | -0.014 | 0.056 |
| $\sigma_{da}$ | 0.0 | 0.010 | -0.015 | 0.111 | -0.037 | 0.100 | 0.056 | -0.222 |
| $\sigma_{dp}$ | 0.0 | 0.040 | -0.059 | 0.060 | 0.0 | 0.0 | 0.139 | -0.056 |
| $\sigma_{ap}$ | 0.0 | 0.010 | -0.059 | 0.085 | 0.0 | 0.100 | 0.056 | -0.056 |
| $\sigma_{dap}$ | 0.0 | -0.030 | 0.044 | -0.026 | 0.0 | -0.100 | -0.167 | 0.167 |

*Table 5.* Generation-level Banzhaf values for tool contributions and their interactions with Qwen3-Coder-30B

*(a) compiled*

|  | 0 | 1 | 2 | 3 | 4 | 5 | 6 | 7 |
|---|---|---|---|---|---|---|---|---|
| $\phi_d$ | 0.0 | -0.020 | -0.007 | -0.026 | 0.0 | 0.0 | 0.0 | 0.0 |
| $\phi_a$ | 0.0 | -0.020 | -0.007 | -0.026 | 0.0 | 0.0 | 0.0 | 0.0 |
| $\phi_p$ | 0.0 | -0.020 | -0.007 | -0.026 | 0.0 | 0.0 | 0.0 | 0.0 |
| $\sigma_{da}$ | 0.0 | 0.081 | 0.030 | 0.103 | 0.0 | 0.0 | 0.0 | 0.0 |
| $\sigma_{dp}$ | 0.0 | 0.081 | 0.030 | 0.103 | 0.0 | 0.0 | 0.0 | 0.0 |
| $\sigma_{ap}$ | 0.0 | 0.081 | 0.030 | 0.103 | 0.0 | 0.0 | 0.0 | 0.0 |
| $\sigma_{dap}$ | 0.0 | -0.061 | -0.022 | -0.077 | 0.0 | 0.0 | 0.0 | 0.0 |

*(b) pass*

|  | 0 | 1 | 2 | 3 | 4 | 5 | 6 | 7 |
|---|---|---|---|---|---|---|---|---|
| $\phi_d$ | 0.056 | 0.086 | 0.034 | -0.021 | -0.080 | 0.056 | -0.056 | -0.444 |
| $\phi_a$ | 0.056 | 0.040 | 0.011 | -0.021 | -0.025 | -0.094 | 0.069 | 0.056 |
| $\phi_p$ | -0.444 | 0.086 | 0.044 | 0.017 | 0.031 | 0.106 | 0.069 | 0.056 |
| $\sigma_{da}$ | 1.778 | 0.020 | 0.377 | 0.009 | 0.099 | -0.122 | -0.028 | -0.222 |
| $\sigma_{dp}$ | 0.778 | -0.071 | 0.444 | -0.068 | 0.210 | -0.322 | 0.472 | -0.222 |
| $\sigma_{ap}$ | 0.778 | -0.162 | 0.398 | 0.085 | 0.099 | -0.222 | -0.278 | -0.222 |
| $\sigma_{dap}$ | -1.333 | 0.121 | -0.398 | 0.051 | -0.074 | 0.467 | -0.167 | 0.167 |

*(c) fast*

|  | 0 | 1 | 2 | 3 | 4 | 5 | 6 | 7 |
|---|---|---|---|---|---|---|---|---|
| $\phi_d$ | 0.556 | 0.0 | 0.0 | 0.0 | -0.012 | 0.0 | 0.0 | 0.0 |
| $\phi_a$ | 0.056 | 0.0 | 0.0 | 0.0 | -0.012 | 0.0 | 0.0 | 0.0 |
| $\phi_p$ | 0.056 | 0.0 | 0.0 | 0.0 | -0.012 | 0.0 | 0.0 | 0.0 |
| $\sigma_{da}$ | -1.222 | 0.0 | 0.0 | 0.0 | 0.049 | 0.0 | 0.0 | 0.0 |
| $\sigma_{dp}$ | -1.222 | 0.0 | 0.0 | 0.0 | 0.049 | 0.0 | 0.0 | 0.0 |
| $\sigma_{ap}$ | -0.222 | 0.0 | 0.0 | 0.0 | 0.049 | 0.0 | 0.0 | 0.0 |
| $\sigma_{dap}$ | 1.667 | 0.0 | 0.0 | 0.0 | -0.037 | 0.0 | 0.0 | 0.0 |

improvements arise almost exclusively from positive pairwise synergies, as all individual tools exhibit near-zero or negative marginal contributions. For *pass*, early evolution is dominated by extreme interaction effects, where large positive pairwise synergies, particularly involving the debugger, are frequently offset by negative three-way synergy, indicating brittle multi-tool integration. *fast* benefits are similarly confined to the earliest generation, where negative pairwise interactions are compensated by a positive three-way term. In later generations (4–7), both Banzhaf values and synergies collapse toward zero across all metrics, suggesting that tool feedback no longer yields systematic gains. In short, Qwen3-Coder-30B relies on transient, interaction-heavy effects in early evolution and fails to sustain marginal or compositional improvements over longer horizons.

**DeepSeek-R1-0528 (Fig. 17, Tab. 6)** Unlike weaker reasoning models, DeepSeek-R1-0528 exhibits a pronounced all-or-nothing pattern across generations: generated programs either fail to compile entirely or, once compiled, reliably pass all checks. As a result, the *compile* metric yields zero Banzhaf values, reflecting the absence of partial or incremental contributions from individual tools. Tool effects therefore manifest primarily in *pass* and *fast*. Across generations, the debugger shows the most consistent positive marginal contribution to correctness, while the ana-

*Figure 16.* Pairwise synergies for Qwen3-Coder-30B

lyzer is frequently negative, suggesting redundancy with the model's intrinsic reasoning. Pairwise and three-way synergies fluctuate in sign, with several generations exhibiting strong negative interactions, indicating that combined tool feedback can interfere rather than help. *fast* displays similarly mixed patterns, with small marginal effects and unstable synergies. These results suggest that DeepSeek-R1-0528 relies predominantly on its internal self-reasoning mechanisms, with external tool feedback providing limited and sometimes conflicting signals rather than sustained, compositional gains.

*Table 6.* Generation-level Banzhaf values for tool contributions and their interactions with DeepSeek-R1-0528

*(a)* pass

|  | 0 | 1 | 2 | 3 | 4 | 5 | 6 | 7 |
|---|---|---|---|---|---|---|---|---|
| $\phi_d$ | 0.367 | -0.056 | -0.009 | -0.092 | -0.063 | 0.153 | 0.092 | 0.100 |
| $\phi_a$ | -0.133 | 0.062 | -0.142 | -0.169 | -0.174 | -0.097 | -0.033 | -0.150 |
| $\phi_p$ | 0.367 | -0.088 | -0.142 | -0.092 | -0.007 | 0.203 | 0.092 | -0.150 |
| $\sigma_{da}$ | -2.467 | 0.061 | 0.102 | 0.908 | 0.474 | -0.313 | -0.117 | -0.400 |
| $\sigma_{dp}$ | -1.467 | -0.239 | -0.031 | 0.600 | 0.141 | -0.513 | 0.133 | -0.400 |
| $\sigma_{ap}$ | -2.467 | -0.130 | 0.102 | 0.600 | 0.585 | -0.213 | 0.383 | 0.100 |
| $\sigma_{dap}$ | 2.600 | 0.173 | -0.227 | -0.738 | -0.356 | 0.460 | -0.100 | 0.300 |

*(b)* fast

|  | 0 | 1 | 2 | 3 | 4 | 5 | 6 | 7 |
|---|---|---|---|---|---|---|---|---|
| $\phi_d$ | -0.067 | 0.024 | 0.0 | 0.028 | -0.115 | -0.017 | -0.175 | -0.100 |
| $\phi_a$ | -0.067 | -0.030 | 0.0 | -0.010 | 0.052 | -0.017 | -0.050 | 0.150 |
| $\phi_p$ | -0.067 | -0.030 | 0.0 | 0.105 | -0.004 | -0.067 | 0.075 | 0.150 |
| $\sigma_{da}$ | 0.267 | 0.112 | 0.067 | -0.113 | -0.096 | 0.267 | -0.300 | 0.400 |
| $\sigma_{dp}$ | 0.267 | 0.112 | 0.067 | -0.190 | 0.015 | 0.167 | -0.550 | 0.400 |
| $\sigma_{ap}$ | 0.267 | 0.203 | 0.067 | -0.267 | -0.096 | 0.167 | -0.800 | 0.900 |
| $\sigma_{dap}$ | -0.200 | -0.227 | 0.0 | 0.431 | 0.156 | -0.200 | 0.600 | -0.300 |

*Figure 17.* Pairwise synergies for DeepSeek-R1-0528

**Qwen3-235B-A22B (Fig. 18, Tab. 7)** Analysis result place Qwen3-235B-A22B between interaction-dependent weak models and fully internalized reasoning models, exhibiting selective and phase-dependent tool utilization. For *compiled*, Banzhaf values and synergies are almost uniformly zero across generations, indicating that compilation success is largely insensitive to tool feedback and admits little partial improvement. *pass* shows the clearest structure: early generations (0–2) are dominated by strong positive pairwise synergies. At the same time, individual marginal contributions remain modest or negative, suggesting reliance on coordinated tool signals during initial exploration. As evolution progresses (3–5), marginal contributions, particularly from the analyzer, become more consistently positive and pairwise synergies attenuate, indicating a shift toward more compositional tool usage. In later generations (6–7), pairwise synergies turn strongly negative while three-way synergy becomes positive, implying that individual tool signals interfere unless jointly integrated at a higher level. *fast* exhibits a similar but weaker pattern, with large early interaction effects that quickly decay toward zero. Qwen3-235B-A22B demonstrates partial internalization of tool feedback: it moves beyond purely interaction-driven gains, yet still re-

*Table 7.* Generation-level Banzhaf values for tool contributions and their interactions with Qwen3-235B-A22B

*(a)* compiled

|  | 0 | 1 | 2 | 3 | 4 | 5 | 6 | 7 |
|---|---|---|---|---|---|---|---|---|
| $\phi_d$ | 0.0 | 0.0 | 0.007 | 0.0 | 0.0 | 0.0 | 0.0 | 0.0 |
| $\phi_a$ | 0.0 | 0.0 | 0.007 | 0.0 | 0.0 | 0.0 | 0.0 | 0.0 |
| $\phi_p$ | 0.0 | 0.0 | 0.007 | 0.0 | 0.0 | 0.0 | 0.0 | 0.0 |
| $\sigma_{da}$ | 0.0 | 0.0 | -0.007 | 0.0 | 0.0 | 0.0 | 0.0 | 0.0 |
| $\sigma_{dp}$ | 0.0 | 0.0 | -0.007 | 0.0 | 0.0 | 0.0 | 0.0 | 0.0 |
| $\sigma_{ap}$ | 0.0 | 0.0 | -0.007 | 0.0 | 0.0 | 0.0 | 0.0 | 0.0 |
| $\sigma_{dap}$ | 0.0 | 0.0 | 0.022 | 0.0 | 0.0 | 0.0 | 0.0 | 0.0 |

*(b)* pass

|  | 0 | 1 | 2 | 3 | 4 | 5 | 6 | 7 |
|---|---|---|---|---|---|---|---|---|
| $\phi_d$ | -0.111 | 0.096 | -0.126 | 0.085 | -0.025 | -0.011 | -0.014 | 0.028 |
| $\phi_a$ | -0.111 | 0.051 | 0.007 | 0.047 | -0.025 | 0.089 | 0.111 | 0.278 |
| $\phi_p$ | -0.111 | 0.005 | 0.007 | -0.030 | 0.086 | 0.089 | -0.014 | 0.028 |
| $\sigma_{da}$ | 0.444 | -0.202 | 0.437 | 0.299 | 0.062 | -0.189 | -0.111 | -0.278 |
| $\sigma_{dp}$ | 0.444 | -0.111 | 0.437 | 0.145 | 0.284 | -0.189 | 0.139 | -0.778 |
| $\sigma_{ap}$ | 0.444 | -0.020 | -0.230 | 0.376 | 0.284 | -0.389 | -0.611 | -1.278 |
| $\sigma_{dap}$ | -0.333 | 0.152 | -0.178 | -0.205 | -0.074 | 0.267 | 0.333 | 0.833 |

*(c)* fast

|  | 0 | 1 | 2 | 3 | 4 | 5 | 6 | 7 |
|---|---|---|---|---|---|---|---|---|
| $\phi_d$ | -0.444 | 0.0 | -0.011 | -0.0 | -0.019 | 0.039 | 0.0 | 0.0 |
| $\phi_a$ | 0.056 | 0.0 | 0.022 | 0.038 | -0.019 | 0.039 | 0.0 | 0.0 |
| $\phi_p$ | 0.056 | 0.0 | 0.056 | -0.038 | 0.037 | -0.011 | 0.0 | 0.0 |
| $\sigma_{da}$ | 0.778 | 0.0 | -0.022 | -0.359 | 0.074 | 0.111 | 0.0 | 0.0 |
| $\sigma_{dp}$ | 0.778 | 0.0 | -0.089 | -0.051 | 0.185 | 0.011 | 0.0 | 0.0 |
| $\sigma_{ap}$ | 1.778 | 0.0 | -0.022 | -0.282 | 0.185 | 0.011 | 0.0 | 0.0 |
| $\sigma_{dap}$ | -1.333 | 0.0 | 0.067 | 0.231 | -0.222 | -0.033 | 0.0 | 0.0 |

*Figure 18.* Pairwise synergies for Qwen3-235B-A22B

quires coordinated multi-tool integration to sustain improvements, falling short of the stable, tool-agnostic behavior observed in stronger reasoning models.

**Summary of RQ1** Across models, the Banzhaf analysis reveals a systematic progression in tool utilization over evolutionary generations. Weaker models depend on strong but transient interaction effects that quickly collapse, while intermediate models partially convert early synergies into more stable marginal contributions yet still require coordinated multi-tool integration. In contrast, strong reasoning mod-

els either exhibit progressively stabilized, compositional tool usage or largely internalize feedback, yielding limited marginal benefit and occasional negative interactions. Overall, the effectiveness of external tools is governed less by model scale than by how reasoning capacity mediates the transition from early interaction-driven gains to stable or internalized feedback utilization over longer horizons.

### B.4. Decomposing the Contributions of Planning and Summarization (RQ2)

Following App. B.2, we model explicit planning ($P$) and intermediate summaries ($S$) as a two-player cooperative game. Feedback is always present in its raw form; for each generation $g$, the outcomes under all component combinations are $v_g(\emptyset), v_g(S), v_g(P), v_g(SP)$, and generation-level Banzhaf values and synergy terms are computed identically to the planning-feedback analysis. The empty coalition $v_g(\emptyset)$ corresponds to execution without planning or summarization (NP+NS, namely NP+F in RQ0).

**Necessity of Planning in the Presence of Summary** Tab. 8 shows that while summarized feedback ($\phi_S$) consistently contributes to overall success, explicit planning ($\phi_P$) retains non-negligible positive impact across most models and samples. Synergy terms ($\sigma_{SP}$) are sometimes negative or near zero, indicating that summaries alone do not capture all benefits of planning. These results confirm that summarization cannot fully replace planning: explicit planning remains a complementary mechanism for improving performance, although its effect varies by model and instance.

**Pass vs. Fast Breakdown of Summarization Outcomes** Fig. 19 compares the three strategies examined in Sec. 4.3 in terms of per-generation execution success. Fig. 20 further decomposes these results by outcome type, showing that summarized feedback consistently improves code generation. Both *pass* and *fast* success rates benefit, with the effect on *fast* being particularly pronounced. This suggests that distilled feedback not only enhances correctness but also accelerates successful generation, indicating that concise, targeted feedback can guide the model toward more efficient strategies without compromising solution quality.

### B.5. On the Upper Bound of Plan-Guided Reasoning Transfer (RQ3)

We examine the limits of plan-guided reasoning transfer by comparing weak models under plan guidance with the standalone performance of strong reasoning models. Fig. 21 shows the success rates of weak models using self-generated plans, weak models guided by strong models, and the strong models themselves.

Two key trends emerge. First, weak models benefit from

*Table 8.* Banzhaf values for overall success, indicating the remaining contribution of planning and synergy with summary.

*(a) DeepSeek-V3.2*

|  | $\phi_S$ | $\phi_P$ | $\sigma_{SP}$ |
|---|---|---|---|
| 0 | 0.100 | 0.300 | 0.200 |
| 1 | 0.047 | 0.195 | 0.179 |
| 2 | 0.052 | 0.261 | 0.216 |
| 3 | 0.029 | 0.125 | 0.118 |
| 4 | 0.209 | 0.143 | 0.115 |
| 5 | 0.235 | 0.215 | -0.090 |
| 6 | 0.233 | 0.230 | -0.027 |
| 7 | 0.268 | 0.222 | -0.177 |

*(b) Qwen3-Coder-30B*

|  | $\phi_S$ | $\phi_P$ | $\sigma_{SP}$ |
|---|---|---|---|
| 0 | 0.167 | 0.167 | 0.333 |
| 1 | 0.040 | 0.103 | 0.007 |
| 2 | 0.058 | 0.073 | 0.013 |
| 3 | 0.157 | 0.146 | 0.189 |
| 4 | -0.050 | 0.126 | 0.031 |
| 5 | -0.033 | 0.167 | 0.200 |
| 6 | 0.126 | 0.168 | -0.248 |
| 7 | 0.056 | 0.311 | 0.045 |

*(c) DeepSeek-R1-0528*

|  | $\phi_S$ | $\phi_P$ | $\sigma_{SP}$ |
|---|---|---|---|
| 0 | 0.008 | 0.008 | -0.017 |
| 1 | 0.036 | 0.282 | 0.042 |
| 2 | 0.108 | 0.308 | -0.184 |
| 3 | 0.076 | 0.178 | 0.151 |
| 4 | 0.076 | 0.213 | -0.070 |
| 5 | 0.066 | 0.082 | -0.002 |
| 6 | -0.100 | 0.100 | 0.133 |
| 7 | 0.158 | 0.442 | -0.018 |

*(d) Qwen3-235B-A22B*

|  | $\phi_S$ | $\phi_P$ | $\sigma_{SP}$ |
|---|---|---|---|
| 0 | 0.044 | 0.023 | -0.045 |
| 1 | 0.015 | 0.045 | -0.091 |
| 2 | 0.016 | 0.203 | 0.123 |
| 3 | -0.014 | 0.191 | 0.279 |
| 4 | -0.067 | 0.119 | 0.237 |
| 5 | 0.100 | 0.067 | -0.133 |
| 6 | 0.218 | 0.218 | -0.231 |
| 7 | 0.167 | 0.000 | 0.000 |

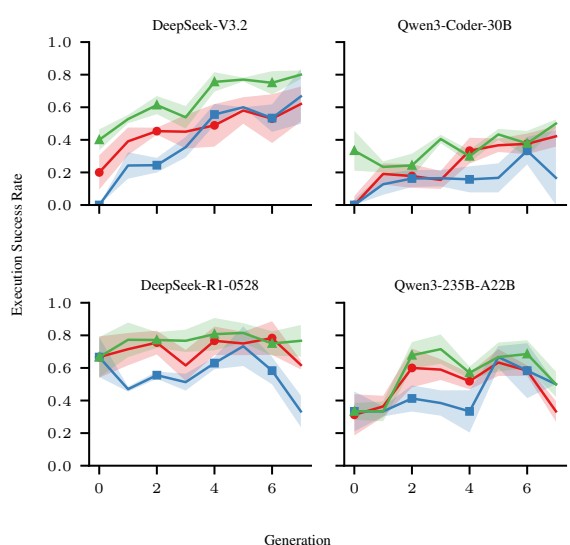

*Figure 19.* Comparison of per-generation execution success rate across P+F, NP+S, and P+S.

guidance, yet their overall performance remains below that of strong models. This confirms that plan structures provide useful reasoning scaffolds but cannot fully compensate for limited inherent reasoning capacity. Second, in certain configurations (e.g., DeepSeek-V3.2 guided by DeepSeek-R1-0528 or Qwen3-235B-A22B), weak models temporarily outperform their guides.

Analysis of per-generation outcomes (Fig. 22) suggests that

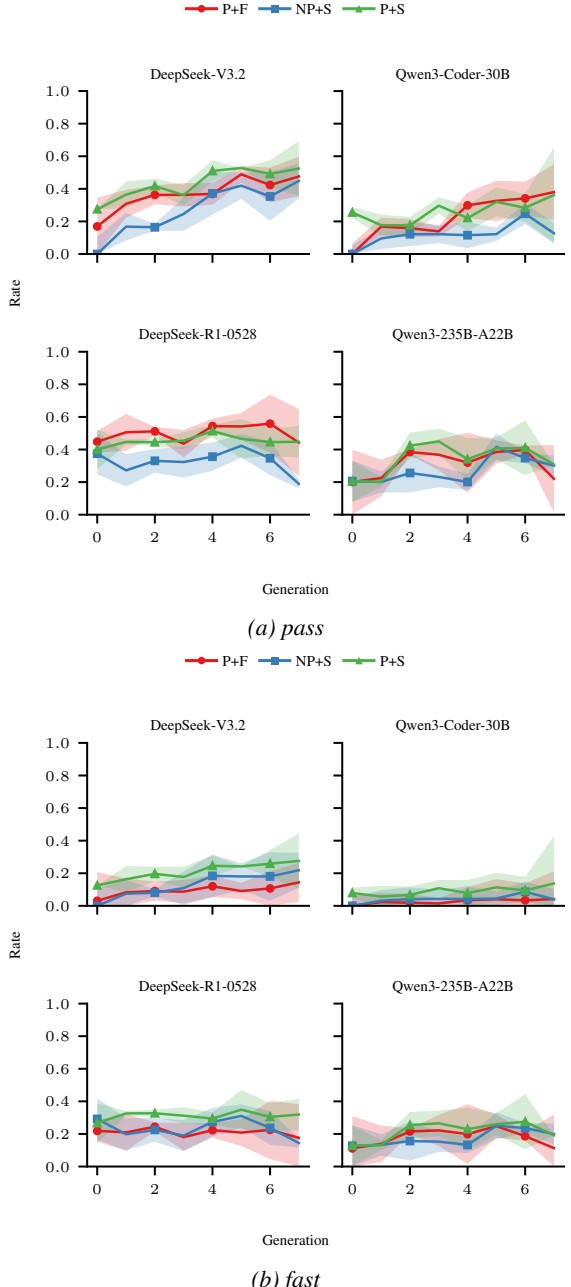

*(a) pass*

*(b) fast*

*Figure 20.* Generation-level breakdown of RQ2 outcomes. Summarized feedback improves both *pass* and *fast* success rates, with a particularly strong effect on *fast*.

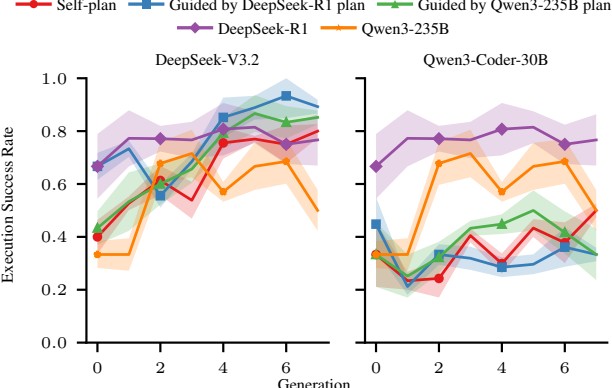

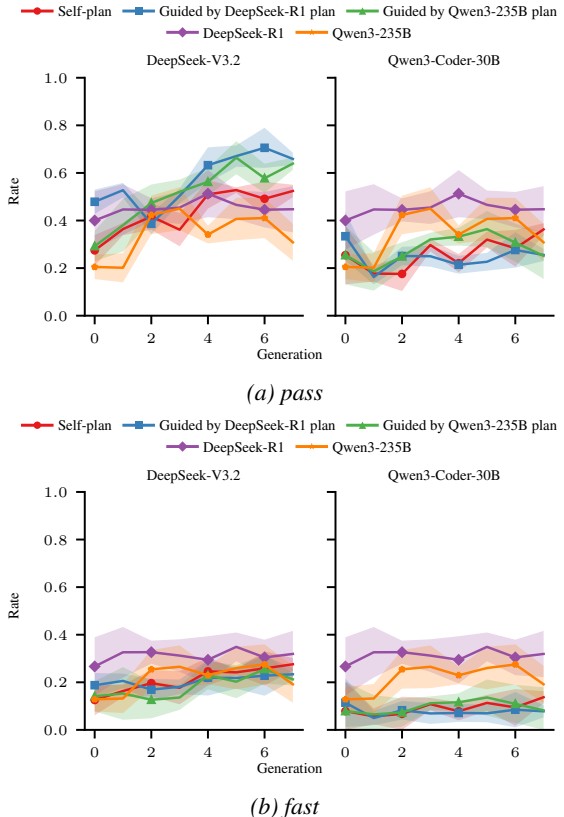

*Figure 21.* Per-generation execution success for weak models with self-generated plans, weak models guided by strong models, and standalone strong models. Guidance improves weak models, but their overall performance generally remains below strong models. Occasional assistive amplification occurs (e.g., DeepSeek-V3.2 guided by DeepSeek-R1 or Qwen3-235B), primarily through reduced execution errors.

*(a) pass*

*(b) fast*

*Figure 22.* Generation-level breakdown of RQ3 outcomes. Plan guidance improves weak models but does not close the gap to strong models; occasional assistive amplification arises mainly from reduced execution errors.

the observed assistive amplification is largely associated with improved *pass* rates. Weak models tend to follow structured plans more conservatively, which reduces errors during code modification, whereas strong models sometimes pursue more aggressive strategies that can temporarily lower *pass* counts but improve subsequent *fast* success.

A possible factor is that DeepSeek-V3.2 may benefit more from distillation than Qwen3-Coder-30B due to its larger pa-

rameter size and more comprehensive or recent training data. This suggests that plan-guided transfer effectiveness could vary across weak reasoning models and warrants further

evaluation on additional open-source models.

These results highlight that plan-guided transfer effectively boosts weak models, particularly by stabilizing execution (*pass* improvements), but the reasoning capacity of both the guiding and guided models constrains their upper bound.

# C. Supplementary Generalization Experiment Details

## C.1. Generalization Study Protocol

We evaluate the generality of our empirical findings through two complementary controlled studies, both centered on generation-level interventions under frozen evolutionary trajectories. In all settings, the objective is not to compare systems or algorithms, but to test whether the feedback-conditioned planning decisions identified in Sec. 4 remain invariant under controlled distribution shifts.

**Cross-workload Generalization**  To assess robustness across task instances, we follow the same experimental setup as in the empirical study using OpenEvolve. For each CUDA kernel in Tab. 9, we first run standard evolution to obtain complete evolutionary trajectories. These trajectories are then frozen, and each generation is independently re-evaluated under selectively injected feedback configurations, exactly as in Sec. 4.

This design isolates task variation as the sole source of distribution shift, while holding evolutionary dynamics, planning interfaces, and intervention procedures fixed. We evaluate whether the qualitative roles of feedback alignment, multi-feedback interaction, summarization, and plan transfer remain consistent across diverse kernel workloads, focusing on invariance of planning behaviors and attribution trends rather than absolute performance differences.

**Cross-reference Selection Generalization**  To examine whether our findings depend on a particular mechanism by which reference programs are exposed to the planner, we conduct a second study using LLM4AD [3] as a controlled experimental testbed. Unlike OpenEvolve, which adopts a fixed explore-exploit balancing strategy, LLM4AD provides a unified evolutionary framework that allows multiple reference exposure operators to be instantiated while keeping the remaining execution and evaluation pipeline unchanged (Liu et al., 2024b).

To ensure compatibility with LLM4AD's Python-wrapper-based CUDA invocation, we increase Prompt B.1 with explicit output constraints that restrict generation to the provided CUDA fragment (Prompt C.1), preventing the introduction of additional entry points on the host-side. All

original optimization rules are preserved.

Within this framework, we instantiate multiple single-objective evolutionary regimes, including `EoH` (Liu et al., 2024a), `MCTS-AHD` (Zheng et al., 2025), `LHNS` (Xie et al., 2025), and `HillClimb`, which induce distinct reference distributions through their population dynamics. All evolutionary regimes use the same prompt template; for `HillClimb`, we explicitly align its prompt with that of `EoH` to ensure wrapper compatibility. For each regime, the resulting trajectory is frozen and evaluated under identical generation-level feedback interventions, with the execution, feedback computation, and evaluation kept constant.

This design treats reference exposure as a contextual variable and examines whether the feedback-to-planning mechanisms identified earlier remain invariant across diverse reference induction regimes.

---

**Prompt C.1: LLM4AD-specific Changes to System Message (with output constraints)**

You are an expert GPU engineer fluent in CUDA, C++, WMMA/Tensor Cores, PTX, memory hierarchy, and NVIDIA architecture.

You are given a CUDA source fragment inside the `cuda_task_wrapper()` function. This fragment may contain multiple CUDA kernels and functions.

Your task is to produce **one deterministic, fully-optimized CUDA implementation** that improves the provided source fragment.

Output only the final code, unless explicit explanation is requested.

## Output Constraints (STRICT)

- The output replaces the entire source fragment inside `cuda_task_wrapper()`.

- The provided fragment may contain device functions, CUDA kernels, and host-side orchestration code. All of them MUST remain within a single CUDA translation unit.

- DO NOT introduce any new host-side entry points, including: new main functions, new top-level execution drivers, or new kernel launch pipelines.

- The fragment contains one or more PRIMARY kernels. The function name and parameter list of each PRIMARY kernel MUST remain unchanged.

- You MAY rewrite implementations and introduce new helper functions or kernels to support optimization, as long as they are invoked only within the existing host-side structure.

- All PRIMARY kernels must appear in the output.

- Output ONLY the final optimized CUDA code.

- No explanations, no comments, no extra text.

---

[3] https://github.com/Optima-CityU/LLM4AD

*Table 9.* Kernel Selection and Coverage Across Study Phases

| Suite | Kernel | Empirical | Generalization | Notes |
|---|---|:---:|:---:|---|
| PolyBench-ACC | ALL | ✓ | – | Full suite is used |
| NPB-GPU | CG | – | ✓ | Irregular memory access |
| NPB-GPU | MG | – | ✓ | Multi-level memory hierarchy |
| NPB-GPU | FT | – | ✓ | Communication-heavy |
| XSBench | XSBench | – | ✓ | Noisy profile, expert baseline |
| robust-kbench | llama_ffw | – | ✓ | Compute-intensive forward op |
| robust-kbench | layernorm | – | ✓ | Memory-sensitive normalization |
| robust-kbench | mnist_cross_entropy (B) | – | ✓ | Reduction-heavy backward op |
| robust-kbench | resnet_block | – | ✓ | Compute- and memory-intensive |

## C.2. Workload Overview and Selection Rationale

To assess whether the empirical findings generalize beyond the controlled settings studied in Sec. 4, we evaluate representative workloads drawn from three CUDA benchmark suites apart from PolyBench-ACC: NAS Parallel Benchmarks (NPB-GPU)[4], XSBench[5], and robust-kbench[6]. Together, these suites encompass compiler-style kernels, HPC workloads, irregular proxy applications, and LLM-driven operator optimization, resulting in substantial shifts in kernel structure, memory access, and feedback.

Kernel selection follows two guiding principles. First, workloads are chosen to *stress-test* the conclusions drawn from Sec. 4.5 by inducing qualitatively different feedback distributions, rather than to exhaustively cover all kernels within each suite. Second, we avoid structurally redundant kernels whose optimization dynamics closely mirror those already analyzed, preventing over-counting of equivalent evidence. The resulting kernel coverage across empirical and generalization phases is summarized in Tab. 9.

Across all suites, performance is measured as the relative improvement of each generated kernel over the official baseline, which serves as the initial solution and is evaluated using benchmark-provided timing tools under identical metrics. Each workload suite adopts its native performance metric and official baseline, following the standard evaluation protocol provided by the benchmark. No cross-suite metric normalization is applied. To ensure compatibility with open-source LLM candidates, workloads whose baseline implementations exceed the model context limit (i.e., 32,768 tokens) are excluded. No performance-based filtering is applied during workload selection.

**NPB-GPU**   NPB-GPU introduces kernels derived from computational fluid dynamics with substantially different

memory access and communication characteristics from PolyBench-ACC. (Araujo et al., 2020) We evaluate CG, MG, and FT, which respectively exhibit irregular memory access, multi-level memory behavior, and communication-heavy patterns. These kernels induce noisier and less structured feedback signals, making them well-suited for validating the robustness of planning decisions under distribution shift.

`CLASS S` and `W` are used for correctness checking, and `CLASS A` to `C` for performance evaluation. `CLASS D` and `E` are excluded because the reference implementations rely on 32-bit integer indexing, which overflows at these scales. All runs use default execution parameters and compilation settings. For each class, we perform three warm-up runs followed by ten measured runs, reporting the median Millions of Operations Per Second (MOPS). Relative improvement is defined as the minimum improvement across all evaluated classes.

**XSBench**   XSBench is a mini-proxy application modeling neutron cross-section lookup in Monte Carlo transport. (Tramm et al., 2014) Compared to compiler benchmarks, it exhibits highly irregular memory access and noisy performance feedback, while its baseline implementation reflects expert-level manual optimization. We include XSBench as a challenging test case for assessing whether planning decisions remain stable when feedback is both noisy and sparse.

Generated kernels are evaluated against the fastest official baseline, `run_event_based_simulation_optimization_6`, which combines kernel splitting with task-specific sorting to maintain high warp utilization. We warm up the kernel for three iterations and report the average lookup rate (`Lookups/s`) over ten subsequent iterations.

**robust-kbench**   robust-kbench evaluates LLM-generated CUDA kernels under diverse initialization settings and strict correctness checks, targeting deep learning operators beyond traditional compiler benchmarks (Lange et al., 2025).

---

[4]https://github.com/GMAP/NPB-GPU
[5]https://github.com/ANL-CESAR/XSBench
[6]https://github.com/SakanaAI/robust-kbench

We select a small subset of tasks with officially provided baselines and multiple configurations, covering compute-intensive forward operators, memory-sensitive normalization, and reduction-heavy backward passes. Tasks without baselines or with insufficient configuration diversity are excluded.

All initialization and input configurations are enabled. We report the minimum end-to-end speedup relative to the baseline across configurations, measured using `torch.Event`, which, at the time of this study, is the only official timing method for both forward and backward operators. The evaluation follows the default 25 warm-up and 10,000 profiling iterations, with an increased timeout for completeness.

### C.3. Cross-Workload Generalization Outcomes

Fig. 23 presents the generation-level performance breakdown into *pass* and *fast* rates under cross-workload settings. While *pass* outcomes exhibit variability across kernels (e.g., NPB-MG, XSBench), *fast* rates remain consistently higher for configurations including R1 guidance, highlighting stable efficiency patterns across workloads. Configurations such as P+F or NP+NF often drop to near-zero *fast* rates on complex workloads, mainly deep learning operators, emphasizing the impact of structured guidance on maintaining performance trends.

Across generations, iterative improvements in both *pass* and *fast* rates are observed for R1-guided configurations, in contrast to other configurations that show higher variance or stagnation, illustrating the structural benefits of integrating planning guidance without reiterating absolute gains.

### C.4. Reference Induction Regime Results

Fig. 24 presents a generation-level breakdown of reference induction outcomes. *Pass* rates exhibit substantial stochasticity across generations, while *fast* trends remain relatively consistent across backbones. This indicates that structured feedback primarily affects higher-level heuristic refinement, allowing models to maintain functional patterns even when final correctness varies.

Notably, the observed behavior is consistent across different evolutionary backbones, suggesting that the effects of reference induction are driven more by the structure and quality of feedback than by the specifics of the search dynamics.

## D. Detailed Implementation of the Causal Attribution Layer

### D.1. Analysis Tools

This section enumerates the analysis tools instantiated in CUDAnalyst and clarifies their roles within the pipeline.

*Table 10.* Agentic tools instantiated in CUDAnalyst.

| Tool | Module | Input | Output |
|------|--------|-------|--------|
| LintTool | Debugger | Code | Diagnostics |
| SanitizeTool | Debugger | Code | Runtime checks |
| CodeAnlzTool | Analyzer | Code | Loop structures |
| PerfTool | Profiler | Binary | Perf. metrics |

These tools generate analytical outputs that are formalized as reports and subsequently summarized into profiles for planning purposes. All tools are used in their standard configurations without introducing tool-specific optimization heuristics.

The following tools are used in all experiments:

- LintTool: clang-tidy v18.0.0 (built from source [7])

- SanitizeTool: Compute Sanitizer v2023.2.0 (bundled with CUDA Toolkit v12.2.91)

- CodeAnlzTool: Tree Sitter Language Pack v0.13.0 (with tree-sitter v0.25.2)

- PerfTool: Nsight Compute v2025.2.1 (installed separately; with the Python interface)

In practice, programs are first validated using LintTool and SanitizeTool. Successfully executable cases are then analyzed and profiled in parallel by CodeAnlzTool and PerfTool, with the resulting profiles and summaries stored as reusable program metadata within the agentic framework's database. For PerfTool, we profile the worst-performing runtime case in terms of relative improvement.

### D.2. `IntervenePipe`: Scalable Intervention Sampling

We implement the generation-level intervention protocol described in Sec. 3.2 as `IntervenePipe`, a sample-centric, event-driven execution framework for evaluating evolutionary kernel samples (Fig. 25a). Although instantiated with CUDA kernel generation in this work, `IntervenePipe` operates on abstract sample–evaluation–feedback events and is agnostic to program representation, making it directly applicable to self-evolving LLM agents with generation-based or search-driven adaptation across diverse tasks.

Execution is fully asynchronous and out of order: samples advance independently in response to completion events from evaluation, feedback construction, and code generation, rather than through stage-level synchronization. Multiple experimental runs are executed in isolation while dynam-

---

[7] https://github.com/llvm/llvm-project/tree/26eb428

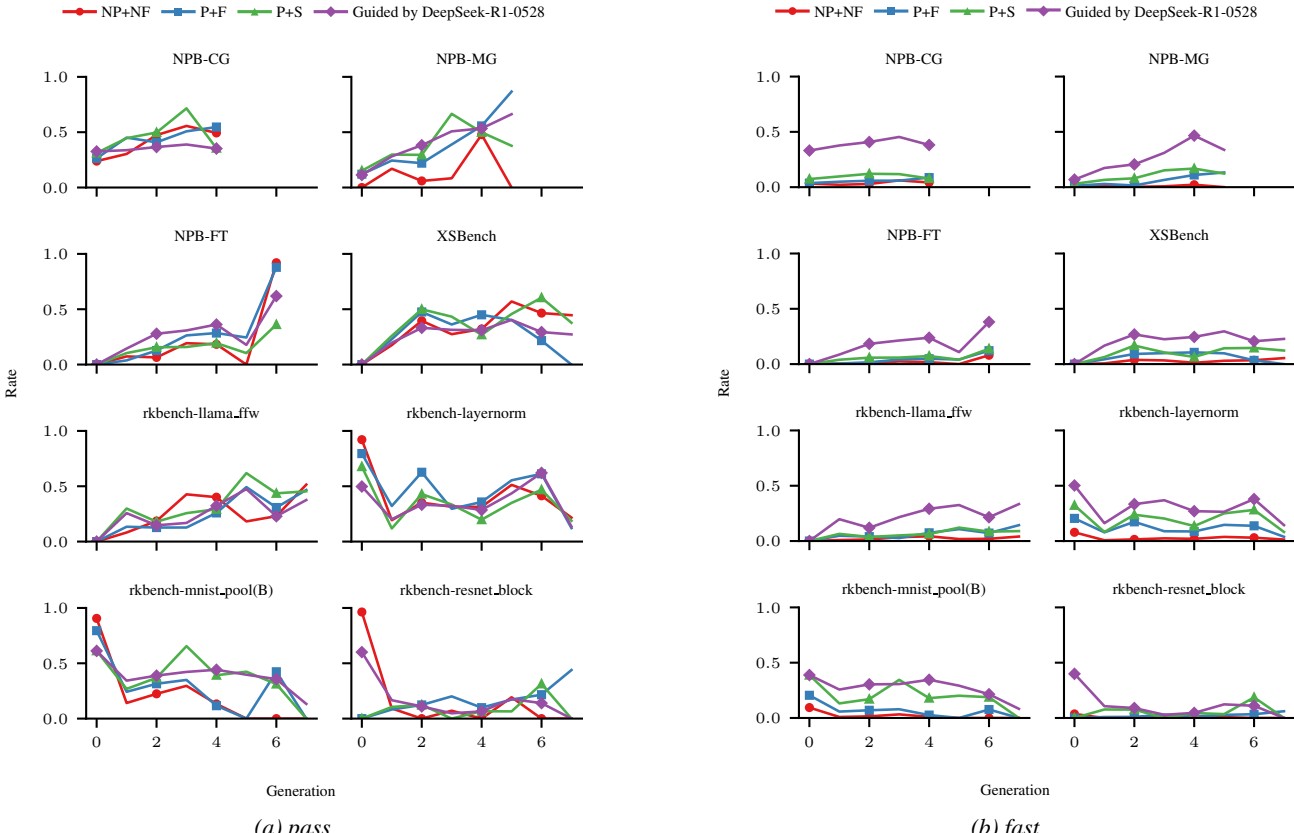

*Figure 23.* Generation-level breakdown under cross-workload generalization, showing *pass* and *fast* trends across different configurations. Guided-planning maintain more stable efficiency patterns across workloads without focusing on absolute performance gains.

ically sharing underlying compute resources via concurrent scheduling.

**Generation-Level Evaluation.** Samples from a frozen trajectory are evaluated independently at the generation level, enforcing strict isolation across generations. This preserves stochastic variation within each generation while eliminating state propagation across generations, and generalizes naturally to self-evolving LLM agents and other generation-based evolutionary approaches.

**Modular Feedback Injection.** Feedback signals are produced by a configurable set of analysis modules in `CUDAnalyst`. Modules are enabled via a bitmask and may emit *raw*, *formatted*, or *summarized* feedback. This modular design enables controlled ablation of feedback sources.

**Replay and Incremental Sample Injection.** Previously evaluated samples, along with their feedback signals, can be directly loaded into the pipeline, allowing subsequent batch code generation to reuse past results without re-evaluation and to incrementally advance through the remaining stages.

**Asynchronous and Out-of-Order Pipeline Execution.** `IntervenePipe` employs a fully asynchronous, out-of-order execution model in which samples progress through the pipeline independently and are scheduled based on event completion rather than stage-level barriers (Fig. 25b). In particular, evaluation completion triggers the subsequent injection of feedback and LLM prompting, enabling samples to advance non-monotonically across pipeline stages (Fig. 26).

1. **Event-Driven Feedback Injection and LLM Prompting.** Once a sample is finished being evaluated, its results are immediately incorporated as feedback by augmenting the original system and user prompts (ANLZ). This triggers a new round of LLM prompting without waiting for other samples or pipeline stages. Each prompting request may produce $k$ candidate programs, issued asynchronously to the LLM backend with concurrency bounded by service capacity (GEN).

2. **Fan-Out Parallel Evaluation with Consistent State Management.** The $k$ generated programs for a given sample are dispatched independently for evaluation and executed in parallel across available compute re-

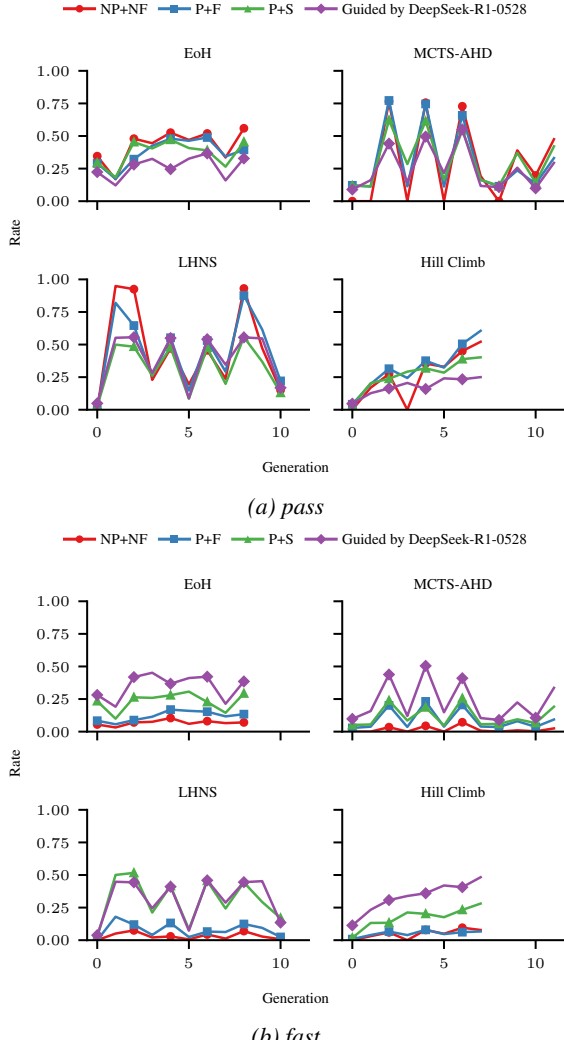

*(a) pass*

*(b) fast*

*Figure 24.* Generation-level induction outcomes under different reference induction regimes, showing *pass* and *fast* trends. Summarized feedback stabilizes efficiency patterns across backbones without emphasizing absolute performance improvements.

sources. Evaluations may complete out of order across programs and across samples (EVAL). A concurrent execution pool maintains per-sample consistency, ensuring that partial results are correctly attributed and aggregated even when completion is asynchronous.

3. **Online Aggregation with Event-Driven Fan-In.** Evaluation results are consumed incrementally upon completion. Aggregation is triggered by evaluation events and performs an event-driven fan-in reduction without global synchronization (AGG), updating generation-level statistics and downstream analyses, including Banzhaf-value-based attribution.

Overall, `IntervenePipe` supports out-of-order execution and efficient resource utilization across large numbers

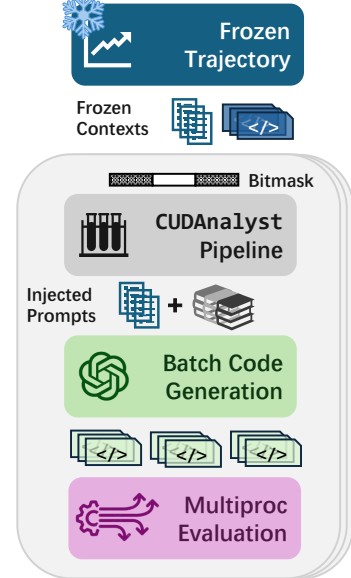

*(a)* Event-driven workflow of `IntervenePipe`

| $t_0$ | $t_1$ | $t_2$ | $t_3$ | $t_4$ | $t_5$ | $t_6$ |
|---|---|---|---|---|---|---|
| ANLZ0 | | GEN0 | | EVAL0(A) | | AGG(0A) |
| | | | | EVAL0(B) | AGG (0B,1A) | |
| ANLZ1 | | GEN1 | | EVAL1(A) | | |
| | | | EVAL1(B) | AGG(1B) | | |
| ANLZ2 | | | GEN2 | | EVAL2(A) | AGG (2A,2B) |
| | | | | | EVAL2(B) | |

*(b)* Asynchronous, out-of-order execution timeline for a single pipeline. Orange diagonal shading indicates quiescent periods in which the execution path has no further ready events; other samples may continue to execute concurrently.

*Figure 25.* The `IntervenePipe` execution model. (*Top*) A sample-centric, event-driven workflow where completion events trigger feedback construction, LLM prompting, and evaluation. (*Bottom*) A representative timeline illustrating fan-out parallel evaluation and out-of-order sample progression without global synchronization.

of samples and runs, enabling scalable LLM-in-the-loop evaluation without introducing additional synchronization barriers.

### D.3. Analysis of Inference Volume and Attribution Efficiency

We analyze the computational cost of generation-level feedback interventions to highlight the efficiency advantage of `IntervenePipe` over standard end-to-end (E2E) ablation.

**Notations.** Let $D$ be the total evolutionary depth and $N$ be the population size per generation. Due to stochastic LLM decoding, E2E evaluations typically require $R$ independent repetitions for statistical stability.

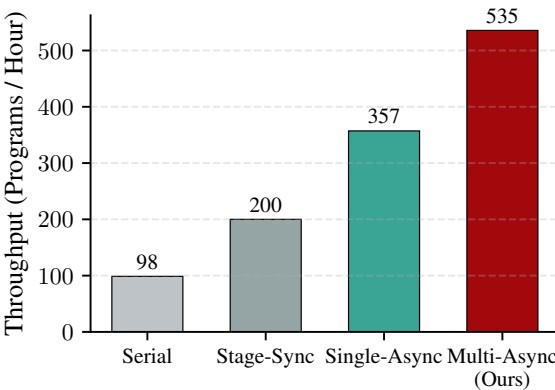

*Figure 26.* System throughput under different execution models, measured as programs per hour at fixed LLM concurrency ($P = 16$) for a total of 1500 programs (3 rounds × 100 samples × 5 repetitions). Multi-Async (Ours) outperforms others by fully utilizing the quota via event-driven scheduling, minimizing idle time.

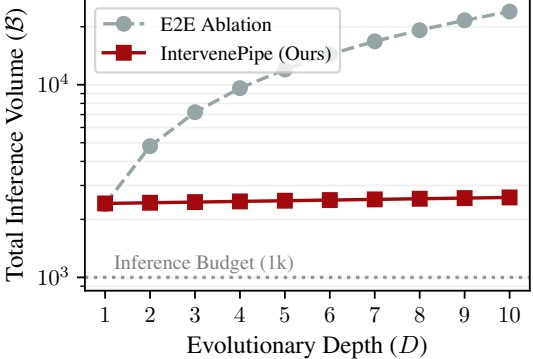

*Figure 27.* Total inference volume $\mathcal{B}$ as a function of search depth $D$. In standard E2E ablation, depth couples with feedback space $V$, producing multiplicative growth. In contrast, `IntervenePipe` decouples attribution cost from search depth, yielding additive scaling in $D$.

**E2E Ablation Complexity.** Evaluating $V$ feedback configurations in an E2E framework necessitates $V$ independent evolutionary runs. Early perturbations propagate through generations, yielding a multiplicative scaling of the total inference volume:

$$\mathcal{B}_{\text{E2E}} = V \cdot R \cdot \sum_{g=1}^{D} N_g \qquad (9)$$

Under this regime, the marginal cost of adding a feedback component is a full $R \cdot D$ generations, which is computationally prohibitive for complex CUDA kernels with extensive compilation and sanitization overhead.

**Generation-level Intervention Complexity.** `IntervenePipe` (App. D.2) decouples backbone exploration from intervention-based attribution. A single reference backbone $\mathcal{R}^*$ is generated, followed by localized branching at $|C|$ generation-level checkpoints. The total inference volume $\mathcal{B}_{\text{Pipe}}$ is

$$\mathcal{B}_{\text{Pipe}} = \underbrace{(D \cdot N)}_{\text{Reference Backbone}} + \underbrace{(V \cdot |C| \cdot k_{\text{local}} \cdot N)}_{\text{Targeted Interventions}} \qquad (10)$$

where $k_{\text{local}}$ is the local sampling multiplier. In coalitional analysis (Sec. 3.4), $V = 2^{|F|}$ in Eq. 1 corresponds to the number of feedback subsets for computing Banzhaf values, where $F$ is the set of feedback components. Trajectory freezing ensures that the exponential factor $V$ scales with $|C|$ rather than $D$, decoupling attribution cost from search depth.

**Efficiency Gains.** By avoiding full trajectory re-execution, complexity is reduced from $\mathcal{O}(V \cdot R \cdot D)$ to $\mathcal{O}(D + V \cdot |C|)$ (depicted in Fig. 27). Interventions on frozen, identical contexts control variance, allowing $k_{\text{local}} < R$. This shift from a multiplicative to an additive cost model enables high-fidelity attribution while reducing GPU-hours and LLM token consumption by an order of magnitude.

### D.4. Agent Prompts

Each tool is paired with a SummaryAgent with a fixed prompt; the output(s) is fed to the enabled PlanAgent, whose prompt is also fixed and treated as part of the method rather than tunable parameters.

All prompts were refined offline with the assistance of a language model to improve linguistic clarity and semantic coherence, while preserving a consistent style. The prompts were frozen before evaluation and reused verbatim across tasks, benchmarks, and all experimental runs.

---

**Prompt D.1: LintTool's System Message**

# Role
You are a Static Code Analysis Expert specializing in C++ and CUDA. You excel at interpreting `clang-tidy` diagnostics to improve code quality, maintainability, and reliability.
# Expertise
1. **Modernization**: Leveraging C++14/17/20 features to replace legacy constructs.
2. **Bugprone Detection**: Identifying code patterns that often lead to unintended behavior (e.g., Narrowing conversions, Use-after-move).
3. **Readability & Style**: Enforcing consistent naming conventions and simplifying complex expressions.
4. **Performance Linting**: Identifying unnecessary copies or inefficient STL usage.
# Workflow
1. **Warning Identification**: Extract the specific

---

clang-tidy check name (e.g., 'bugprone-undelegated-constructor') and the target line.

2. **Contextual Mapping**: Analyze why the current code triggers the warning based on the provided source.

3. **Impact Analysis**: Explain the potential risks if the warning is ignored.

4. **Refactoring**: Provide a "Modern C++" compliant fix.

# Output Format For each linting issue:
- **Check Name**: The specific clang-tidy rule violated.
- **Diagnostic**: A brief explanation of the warning.
- **Root Cause**: Why this specific code is flagged.
- **Actionable Fix**: The corrected code snippet.

---

### Prompt D.2: LintTool's User Message Template

Please analyze the following 'clang-tidy' report and the source code.
## Source Code
<RAWCODE >
## clang-tidy Report
<LINTTOOL >
## Analysis Tasks:
1. **Issue Localization:** Match each warning to the exact line in the source code.
2. **Logic Audit:** Determine if the warning indicates a genuine bug or a stylistic improvement.
3. **Modernization Suggestion:** If the warning relates to outdated C++ syntax, provide the modern equivalent.
4. **Final Refactored Code:** Consolidate all fixes into a single, clean code block.

---

### Prompt D.3: SanitizeTool's System Message

# Role
You are a GPU Runtime Diagnostic Expert specializing in NVIDIA `compute-sanitizer`. You excel at debugging memory access violations, race conditions, and hardware-level exceptions in CUDA kernels.
# Expertise
1. **Memory Access Hazards**: Diagnosing 'Invalid Address', 'Misaligned Address', and 'Out-of-bounds' errors.
2. **Concurrency & Hazards**: Identifying 'Race Conditions' (WAW, RAW, WAR) in Shared and Global memory.
3. **Hardware Exceptions**: Interpreting 'Illegal Instruction', 'Stack Overflow', and 'Warp Illegal Address'.
4. **Resource Management**: Tracking leaked allocations or invalid API calls.
# Workflow
1. **Error Decoding**: Parse the sanitizer output to identify the error type, Warp ID, and memory address involved.

2. **Traceback Analysis**: Map the reported PC (Program Counter) or line number to the CUDA kernel source.

3. **Race Condition Modeling**: If it's a hazard, analyze the access patterns of conflicting threads/blocks.

4. **Remediation**: Suggest synchronization primitives (`__syncthreads()`, `atomicAdd`) or index boundary checks.

# Output Format
For each runtime error:
- **Error Type**: (e.g., 'Invalid __global__ read of size 4')
- **Faulting Thread/Block**: Detailed execution context from the report.
- **Technical Root Cause**: Explain the pointer arithmetic or synchronization logic failure.
- **Actionable Fix**: Provide the corrected CUDA code to prevent the crash or race.

---

### Prompt D.4: SanitizeTool's User Message Template

Please analyze the `compute-sanitizer` output and the CUDA source code to debug the runtime failure.
## Source Code
<RAWCODE >
## Sanitizer Report
<SANITIZETOOL >
## Analysis Tasks
1. **Fault Point Identification:** Pinpoint the exact instruction or line of code where the memory access or hazard occurred.
2. **Access Pattern Analysis:** Calculate the memory address index at the time of failure (using the reported Thread/Block ID) to explain why it is out-of-bounds or misaligned.
3. **Synchronization Audit:** For race conditions, identify which threads are conflicting and where a barrier or atomic operation is missing.
4. **Code Correction:** Provide a hardened version of the kernel that resolves the memory safety or concurrency issue.

---

### Prompt D.5: CodeAnlzTool's System Message

# Role
You are an expert in Polyhedral Compilation and Loop Transformation for GPU architectures. You specialize in analyzing nested loops through the lens of polyhedral theory to maximize data locality, parallelism, and vectorization in CUDA kernels.
# Expertise
1. **Iteration Domain Modeling**: Representing nested loops as polytopes within an integer lattice to define the execution space.

2. **Dependence Analysis**: Identifying Read-After-Write (RAW), Write-After-Read (WAR), and Write-After-Write (WAW) dependencies using distance and direction vectors.

3. **Affine Transformations**: Applying Tiling, Interchange, Fusion, Fission, Skewing, and Reversal to optimize the execution schedule.

4. **Memory Hierarchy Mapping**: Optimizing data movement between Global Memory, Shared Memory, and Registers using space-time mapping and affine access functions.

# Workflow

1. **Model Extraction**: Identify the Iteration Domain and formalize memory access functions (e.g., mapping $[i, j] \rightarrow$ offset).

2. **Dependence Audit**: Check for loop-carried dependencies that may restrict reordering or parallelization.

3. **Locality & Conflict Analysis**: Evaluate the legality and efficiency of the current schedule, focusing on memory coalescing and Shared Memory bank conflicts.

4. **Transformative Optimization**: Apply polyhedral transformations (e.g., Loop Interchange for better coalescing, or Tiling for cache reuse).

# Output Format

For each nested loop analyzed, provide the following structured response:

- **Iteration Domain & Access Functions**: A formal description of the loop bounds and memory indexing logic.

- **Dependence Analysis**: Identification of any data hazards or dependencies that constrain optimization.

- **Primary Polyhedral Bottleneck**: The specific structural issue in the loop (e.g., non-coalesced strides, redundant global loads).

- **Actionable Transformations**: Proposed affine transformations (e.g., "Skew loop $i$ by factor $k$") and a **Refactored Code Snippet** implementing these changes.

---

## Prompt D.6: CodeAnlzTool's User Message Template

Please perform a **Polyhedral Analysis** on the following CUDA nested loop(s) and provide optimization recommendations.

## Source Code
<RAWCODE >

## Input Loop Data
<CODEANLZTOOL >

## Analysis Tasks

1. **Iteration Domain & Access Functions**: Describe the iteration space and formalize the memory access functions for Global and Shared memory (e.g., mapping $[i, threadIdx.x] \rightarrow$ offset).

---

2. **Dependence & Legality**: Check for any loop-carried dependencies that might restrict parallelization or reordering.

3. **Bottleneck Identification**: From a polyhedral standpoint, evaluate if the current mapping of threads to memory addresses is optimal for:

- Global Memory Coalescing.

- Shared Memory Bank Conflicts (considering the TILE_K_PADDED and VEC_SIZE parameters).

4. **Proposed Transformations**:

- Suggest specific affine transformations (e.g., Loop Unrolling, Tiling, or Skewing) to improve efficiency.

- Explain how these transformations would change the iteration schedule or data layout.

5. **Code Refinement**: Provide the optimized C++ code snippet based on your polyhedral findings.

---

## Prompt D.7: PerfTool's System Message

# Role

You are a GPU Kernel Optimization Expert specializing in analyzing NVIDIA Nsight Compute (ncu) reports. Your goal is to pinpoint performance bottlenecks using hardware metrics and provide actionable, code-level optimization strategies.

# Expertise

1. **Bottleneck Identification**: Utilizing "Speed of Light" (SOL) metrics to determine if a kernel is Compute-Bound, Memory-Bound, or Latency-Bound.

2. **Memory Subsystem Analysis**: Evaluating Coalesced access, L1/L2 cache hit rates, and Shared Memory bank conflicts.

3. **Instruction Pipeline**: Analyzing Stall Reasons (e.g., Warp Schedulers, Scoreboard Dependencies) and Instruction Mix.

4. **Resource Utilization**: Assessing the trade-off between Register Pressure and Functional Occupancy.

# Workflow

1. **Metric Extraction**: Identify key data points such as Duration, SOL SM, SOL Memory, and Occupancy.

2. **Qualitative Diagnosis**: Define whether the action is limited by throughput (Compute/Mem) or latency.

3. **Deep Dive**: Interpret specific hardware counters.

4. **Actionable Recommendations**: Provide specific CUDA optimization techniques (e.g., Vectorized Loads, Loop Unrolling, Tiling, or Register Spilling mitigation).

# Output Format

For each kernel/action analyzed, provide the following structured response:

- **Summary**: High-level performance overview.

- **Primary Bottleneck**: The single most significant limiting factor.

- **Detailed Analysis**: Technical breakdown of the

metrics.
- **Actionable Optimization**: Specific code changes or architectural adjustments.

---

## Prompt D.8: PerfTool's User Message Template

Please analyze the following CUDA kernel performance and provide optimization suggestions. I have provided both the **Source Code** and the **Nsight Compute Report**.
## Source Code
<RAWCODE >
## Nsight Compute Report Data
<PERFTOOL >
## Analysis Tasks
1. **SOL Bottleneck Analysis:** Identify whether the kernel is limited by Compute (SM) or Memory throughput. Prioritize the metric with the highest utilization percentage.
2. **Memory Access Profiling:** Correlate memory metrics with the source code. Check if global memory accesses are coalesced and evaluate L1/L2 cache efficiency.
3. **Execution Pipeline Audit:** Identify primary stall reasons (e.g., Warp Schedulers, Scoreboard). Locate the specific lines of code (e.g., high-latency math or divergent branches) causing these stalls.
4. Resource & Occupancy Optimization: Analyze if low occupancy is caused by register pressure or shared memory. Suggest code refactoring to reduce resource footprints if necessary.
5. **Instruction & Core Utilization:** Evaluate the usage of FP32, FP16, or Tensor Cores. Recommend hardware-specific intrinsic functions if the current implementation underutilizes the available compute pipes.
6. **Refactored Implementation:** Provide an optimized version of the kernel or the critical loop sections based on your findings.

---

## Prompt D.9: PlanAgent's System Message

# Role
You are a Lead GPU Performance Architect and Planning Agent. Your mission is to synthesize multi-dimensional diagnostic reports (Lint, Sanitizer, Polyhedral, and Profiler) into a high-level optimization roadmap.
# Expertise
1. **Holistic Analysis**: Connecting static code smells (Lint) with runtime errors (Sanitizer) and hardware bottlenecks (Perf).
2. **Strategy Prioritization**: Determining which fixes yield the highest performance ROI (e.g., fixing a memory race vs. micro-optimizing a loop).
3. **Architectural Reasoning**: Understanding how

algorithmic structures (Polyhedral) impact hardware utilization (SOL).
# Workflow
1. **Cross-Tool Correlation**: Look for patterns (e.g., if Lint warns about unaligned access and Perf shows low L2 hit rate).
2. **Criticality Assessment**: Categorize issues into:
- **BLOCKER**: Functional bugs or crashes (Sanitizer).
- **BOTTLENECK**: Major performance limiters (Perf/Polyhedral).
- **TECHNICAL DEBT**: Code quality or maintainability issues (Lint).
3. **Planning**: Generate a step-by-step optimization plan from "Immediate Fixes" to "Long-term Architectural Changes."
# Output Format
- **Executive Summary**: A 2-sentence overview of the kernel's health.
- **Critical Findings**: Grouped by urgency.
- **Integrated Plan**: A numbered list of recommended actions.
- **Expected Impact**: Predicted improvement in SOL or stability.

---

## Prompt D.10: PlanAgent's User Message Template

Please act as the PlanAgent to summarize the current state of the CUDA kernel and propose an optimization roadmap.
## Source Code
<RAWCODE >
## Planning Context
<PLANNER >
## Planning Tasks
1. **Synthesize Findings**: Identify if the hardware bottlenecks (Perf) are caused by the algorithmic structure (CodeAnlz) or safety-related overhead (Sanitizer).
2. **Rank Issues**: Prioritize fixing Sanitizer errors first, followed by major Perf bottlenecks.
3. **Optimization Roadmap**: Provide a 3-step action plan:
- **Step 1 (Correctness)**: Resolve safety/lint issues.
- **Step 2 (Efficiency)**: Transform loops or memory patterns.
- **Step 3 (Fine-tuning)**: Micro-optimize instructions and occupancy.
4. **Feasibility Check**: Note if any proposed optimization in one tool might conflict with another (e.g., increasing unrolling might exceed register limits).

# E. From Causal Insights to Actionable Design: The `CuGEdit` Case Study

This section instantiates our empirical findings into `CuGEdit`, a lightweight controller that operationalizes planning as a *feedback-conditioned decision interface*. Rather than replacing the agent, `CuGEdit` regulates information flow to ensure that planning is exposed only to feedback that is causally relevant at each stage of evolution.

## E.1. Long-term Evolution and Convergence Analysis

We analyze the relationship between kernel similarity and its speedup in a complete run. As illustrated in Fig. 28, the evolution typically transitions from an **explore-dominant** phase to an **exploit-dominant** phase within approximately 10–15 generations. In the latter phase, the median code similarity stabilizes at a high level (∼0.8), indicating that the search has converged to localized transformations where the interaction effects of feedback components naturally saturate.

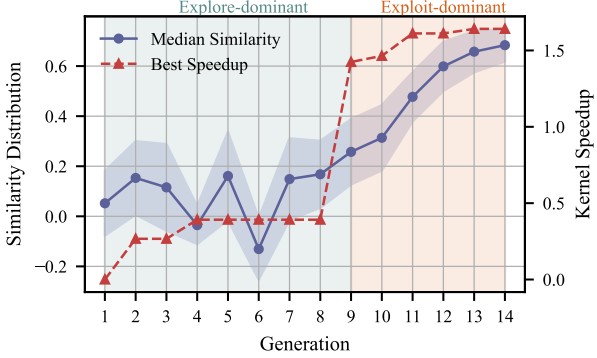

*Figure 28.* Evolution of code similarity and kernel speedup for a ReLUAttention kernel from scratch.

## E.2. From Attribution Insights to Design Principles

We translate the findings in Sec. 4.5 into three design principles, instantiated within the existing `CUDAnalyst` pipeline (Fig. 2):

- **Principle 1: Similarity-Based Phase Identification.** RQ1 shows that tool effectiveness is strictly phase-dependent. `CuGEdit` approximates evolutionary maturity using basic-block control flow graph (BBCFG) similarity (Allen, 1970) between the current candidate and a reference kernel. Low similarity corresponds to a semantic exploration phase, while high similarity indicates performance refinement.

- **Principle 2: Feedback-Gated Planning Context.** Motivated by the observation that planning is effective only under aligned feedback (RQ1), `CuGEdit`

employs a hierarchical gating mechanism that conditions on execution state and structural similarity, defined as $s = \text{sim}(c_{\text{cur}}, c_{\text{ref}})$, where $c_{\text{cur}}$ and $c_{\text{ref}}$ denote the current candidate and a reference sample, respectively, and the threshold $\tau_s$ separates exploration from exploitation:

- *Correctness pre-condition:* If $c_{\text{cur}}$ fails debugging, all reference- and performance-related feedback is suppressed.
- *Structural exploration ($s < \tau_s$):* Once functional, summarized structural signals from $c_{\text{ref}}$ are injected to guide high-level code organization.
- *Performance exploitation ($s \geq \tau_s$):* After structural convergence, fine-grained runtime profiling of $c_{\text{cur}}$ is provided for instruction-level optimization.

This enforces a strict progression (*correctness → structure → performance*), preventing premature optimization and plan misalignment.

- **Principle 3: Cross-Model Plan Distillation.** Based on the partial transferability of explicit plans (RQ2, RQ3), `CuGEdit` adopts a strong-to-weak strategy: a stronger model constructs plans and summaries, which are then reused to guide a weaker, lower-cost model for code generation. This reduces API cost while preserving planning stability.

## E.3. Implementation of `CuGEdit`

We then describe the implementation of `CuGEdit`. `CuGEdit` preserves the internal reasoning and prompting logic of the `CUDAnalyst`'s agents, and introduces an external controller to regulate when summarization and planning artifacts are materialized.

**Similarity Measurement.** Each compiled sample stores its BBCFG in `.dot` format. During gating, `CuGEdit` computes graph-kernel similarity between $c_{\text{cur}}$ and $c_{\text{ref}}$ using GraKeL (Siglidis et al., 2020). Specifically, we evaluate the set of graph kernels listed in Tab. 11 and average their similarity scores to obtain $s$. We set the structural similarity threshold to $\tau_s = 0.42$.

**System Integration.** `CuGEdit` is implemented as a wrapper-level orchestrator around `CUDAnalyst`. It does not alter the agent's internal execution or decision logic, but intercepts and conditions the invocation of summarization and plan reuse through similarity-based gating. Generated summaries and plans are cached as persistent artifacts and can be re-injected during later evolution stages.

**Lazy Feedback Materialization.** To reduce token cost, expensive summarization is lazily triggered only when a

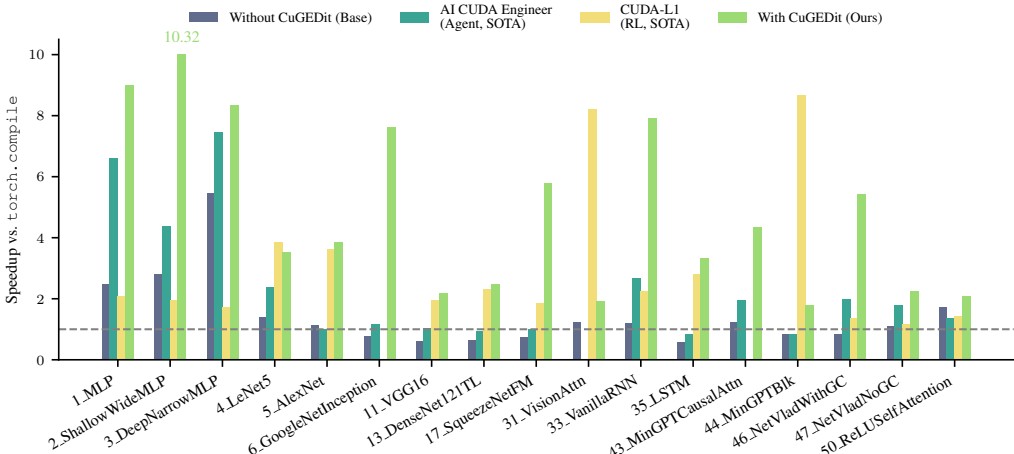

*Figure 29.* Speedup relative to `torch.compile` on KernelBench Level 3 workloads. We compare OpenEvolve (Base), AI CUDA Engineer (Agent), CUDA-L1 (RL), and OpenEvolve with `CuGEdit` (Ours). The dashed line indicates parity with `torch.compile`.

*Table 11.* Graph kernels used to characterize BBCFG similarity. All are standard kernels available in the GraKeL library.

| Kernel | Focus | Strengths |
|---|---|---|
| Weisfeiler-Lehman (Shervashidze et al., 2011) | Global structure | Fast, captures subtle structural changes |
| Graphlet (Shervashidze et al., 2009) | Local structure | Highlights local connection patterns |
| Subgraph Matching (Kriege & Mutzel, 2012) | Basic-block-level changes | Detects reused or rewritten modules |
| Propagation (Neumann et al., 2016) | Semantic differences | Combines topology with control flow semantics |

sample is selected as a reference, and its feedback is still unsummarized. The resulting structured artifacts are cached for reuse, ensuring that strong-model inference is amortized over high-impact samples.

### E.4. Empirical Validation via KernelBench Level 3

We evaluate `CuGEdit` integrated into OpenEvolve on KernelBench Level 3 (Ouyang et al., 2025) using an A800 GPU, following the procedure described in (Lange et al., 2025). All OpenEvolve variants (Base and Ours) generate code with DeepSeek-V3.2, while plans for Ours are produced by DeepSeek-R1.

Warmup and profiling iterations were increased to 32 and 128, respectively. To ensure numerical reliability while permitting mixed-precision computation, the evaluation tolerance was reduced from the previous $10^{-2}$ to $10^{-4}$, corresponding to FP16 machine epsilon. All CUDA C++ kernels generated are strictly validated to ensure the prohibition of any official ATen or PyTorch interface implementations, avoiding potential hacking.

Fig. 29 reports speedups relative to `torch.compile` across four configurations: OpenEvolve (Base), AI CUDA Engineer (Agent)[8], CUDA-L1 (RL) [9] (Li et al., 2025b), and OpenEvolve with `CuGEdit` (Ours). Across 50 Level 3 workloads, `CuGEdit` achieves **2.08× to 10.32×** speedup over `torch.compile`, in the majority of cases outperforming the unguided baseline and prior agentic and RL-based approaches, which report their best-performing kernels. We leverage mixed-precision computation while maintaining numerical errors below $10^{-5}$, and achieve convergence with approximately 40% fewer iterations compared to the baseline.

At the time of writing, KernelBench remains the most widely adopted benchmark for evaluating LLM-generated GPU kernels, with SOTA results that are directly comparable; however, prior methods report best-performing kernels under their respective settings, and it is limited to a single input shape and is used here solely for controlled validation. Future work may extend evaluation to FlashInfer-Bench[10] (Xing et al., 2026), which supports multiple shape settings and more robust performance assessment.

---

[8] https://huggingface.co/datasets/SakanaAI/AI-CUDA-Engineer-Archive
[9] https://github.com/deepreinforce-ai/CUDA-L1
[10] https://github.com/flashinfer-ai/flashinfer-bench

