# OpenReview forum: "Towards Feedback-to-Plan Decisions for Self-Evolving LLM Agents in CUDA Kernel Generation"
_ICML.cc/2026/Conference — ICML 2026 regular_

### Official Review · Reviewer_fYwX · 2026-03-08

**Soundness:** 3
**Presentation:** 2
**Significance:** 3
**Originality:** 3
**Overall Recommendation:** 4
**Confidence:** 3

**Summary:**

The research addresses a central domain of LLM-driven self-evolving agent design for high-performance CUDA kernel generation, and the manuscript intends to examine the challenge of disentangling the attribution and interaction of heterogeneous feedback signals in feedback-to-plan decisions, as well as overcoming the limitations of traditional end-to-end ablation in analyzing such decisions for iterative evolutionary systems.

Designs and implements the CUDAnalyst unified analysis layer, which decouples feedback and planning via trajectory freezing and selective feedback injection, enabling controlled generation-level feedback intervention and fine-grained component attribution, and is agnostic to the underlying agent framework.

Designs the CuGEdit modular plugin based on empirical findings, implementing feedback-to-plan optimization strategies, which achieves improved convergence efficiency on KernelBench, outperforming existing SOTA methods.

**Compliance With Llm Reviewing Policy:**

Affirmed.

**Key Questions For Authors:**

1. The research does not analyze the computational cost and token consumption of CUDAnalyst. For a single kernel generation, how many LLM rollouts are required on average to obtain stable attribution conclusions? Please also supplement the quantitative comparison of CUDAnalyst with traditional end-to-end ablation experiments in terms of LLM inference token count and GPU time consumption.
2. The paper only focuses on the analysis of valid feedback. If noisy or erroneous feedback signals are introduced, can CUDAnalyst identify their negative impacts on planning decisions? Is there a corresponding quantitative evaluation or anti-interference mechanism design?
3. The research mainly analyzes the feedback effects in short-term generations. In ultra-long-term evolution (e.g., more than 20 generations), will the marginal contribution and interaction effect of feedback components decay, saturate or reverse?

**Limitations:**

yes

**Strengths And Weaknesses:**

strengths:
1. Proposes the intervention-based CUDAnalyst framework to address the "trajectory drift" in LLM agent evaluation, providing a new tool for analyzing complex self-evolving systems in CUDA kernel generation.
2. Adopts game theory analysis with Banzhaf values and interaction terms to quantify feedback tool contributions, making evaluation more convincing and interpretable than simple performance percentage analysis.
3. Experimental results show high consistency across multiple workloads and reference induction regimes, verifying the robustness of the conclusions.
4. Implements the CuGEdit plugin based on theoretical insights, translating analytical findings into actual performance improvements with both theoretical and practical values.

weaknesses：
1. Validations only focus on NVIDIA CUDA kernel generation, with no verification of the conclusions' adaptability on GPUs of other architectures such as AMD.
2. No explicit mechanism is designed to introduce professional knowledge and experience in the kernel field, without integrating domain expert capabilities.
3. CUDAnalyst only incorporates three types of feedback (debugger, analyzer, profiler), without considering human feedback, cross-task experience summary and other forms, and lacks memory and experience summary modules.
4. CuGEdit is only validated on KernelBench Level 3, with no analysis of its performance and efficiency in large-scale, industrial-grade CUDA kernel generation scenarios, and the verification of practical application value is still in the initial stage.

---

> ### Author Rebuttal · Authors · 2026-03-31
>
> Additional appendix (F to H): https://anonymous.4open.science/api/repo/additional-appendix-4FB2/file/Additional_Appendix.pdf
>
> > Q1: How many LLM rollouts are required on average to obtain stable attribution conclusions? Please also supplement the quantitative comparison of CUDAnalyst with traditional end-to-end ablation experiments
>
> For each frozen context, we estimate $v(S)$ using $k=5$ samples across 10 independent runs to ensure statistical significance. While our empirical observations show that the localized nature of Trajectory Freezing yields stable estimates with as few as 5 runs, we opted for 10 runs to further bolster the robustness of our findings. Fig. 10 depicts that selecting $k=5$ for PolyBench-ACC empirical evaluation is enough.
>
> This highlights a key advantage of CUDAnalyst: by isolating causal effects from cumulative stochastic drift (**Figure 31**), our framework achieves precise feedback attribution with significantly lower computational overhead than would be required to achieve similar confidence levels in a traditional E2E ablation setting.
>
> In **Appendix D.3**, we explicitly compare the 'Attribution Efficiency' of CUDAnalyst against traditional end-to-end (E2E) ablations. We demonstrate that while E2E requires massive full-trajectory repetitions to overcome the 'Stochastic Drift' (detailed in **Appendix H.1**), CUDAnalyst isolates the immediate causal impact. This "surgical" approach provides much higher attribution clarity with a significantly lower inference budget, avoiding the extensive repetitions typically required to extract statistically significant conclusions from end-to-end (E2E) performance alone.
>
> > Q2: If noisy or erroneous feedback signals are introduced, can CUDAnalyst identify their negative impacts on planning decisions?
>
> Yes. We explicitly addressed this in **Section 4.1 (RQ0)** using "DummyPlan" (DP) and "Randomized Feedback" (RF). When noisy or irrelevant signals were introduced, CUDAnalyst captured a significant drop in planning effectiveness **(Figure 12, Appendix B.2.)**, proving it can effectively diagnose the negative impact of poor-quality feedback.
>
> > Q3: In ultra-long-term evolution (e.g., more than 20 generations), will the marginal contribution and interaction effect of feedback components decay, saturate, or reverse?
>
> We provide an analysis of long-term convergence in **Appendix H.2 (Figure 32)**. The results show that evolution typically transitions from "exploration" to "exploitation" within 10-15 generations for CUDA kernel optimization. Beyond 20 generations, marginal contributions often saturate as code similarity stabilizes at ~0.8, indicating the agent has reached a localized optimal transformation where feedback-driven gains diminish.
>
> > W1: Validations only focus on NVIDIA CUDA kernel generation.
>
> As mentioned in the response to Reviewer RMfA, we have included a **Python JIT kernel generation** study in **Appendix G** to demonstrate cross-language (CUDA C++ to Python) and cross-platform (CUDA GPU to a multicore CPU) generalizability. The analytical primitives in CUDAnalyst (e.g., polyhedral analysis and NCU profiler) have direct functional equivalents in AMD ROCm, as instantiated in [1].
>
> > W2: Without considering human feedback, cross-task experience summary, and other forms, it lacks memory and experience summary modules.
>
> We clarify that CUDAnalyst is specifically designed to **isolate the immediate causal impact** of feedback on planning. To achieve this "surgical" precision, we intentionally **freeze the implicit memory** (represented by Elite/Reference Programs) as a constant background at specific trajectory points. This eliminates the confounding noise of "experience drift," allowing us to resolve how a model decides its next move based strictly on technical diagnostics.
>
> **Empirical validation shown in Figure 9 and Figure 22** demonstrates striking structural invariance across diverse reference distributions generated by varied evolutionary operators (i.e., EoH, MCTS-AHD, LHNS, and hill-climbing). This synchronized and deterministic response proves that the core planning logic is effectively decoupled from the underlying "experience" distribution, ensuring rigorous feedback quantification.
>
> Attributing planning gains to the "experience" embedded in CUDA C++ kernels remains an open challenge in compiler research (namely, the PSEC problem [2]). Therefore, freezing the elite state is a deliberate methodological choice to maintain evaluative purity. This "frozen-trajectory" approach provides a robust and generalizable template for other state-heavy domains where semantic characterization of complex states is non-trivial.
>
> ****
>
> [1] SwizzlePerf: Hardware-Aware LLMs for GPU Kernel Performance Optimization, https://openreview.net/forum?id=a5aJi9OAr0
> [2] Program State Element Characterization, https://dl.acm.org/doi/10.1145/3579990.3580011

---

> > ### Author Rebuttal · Reviewer_fYwX · 2026-04-01
> >
> > Thank you for your rebuttal. The overall reply is clear.
> >
> > 1.Regarding the response to Q1: Could you provide further comparisons on token consumption? Given that CUDAnalyst1.  performs extensive reflection, does it incur high token costs?
> >
> > 2.Regarding the response to Q2: The authors confirm that CUDAnalyst can detect performance drops from noisy feedback. Could you explain the mechanism for adapting or correcting planning decisions under erroneous feedback?
> >
> > 3.Could you briefly summarize the strengths and weaknesses of your approach in comparison with NVIDIA's recent work AVO?

---

> > > ### Author Response · Authors · 2026-04-06
> > >
> > > We thank the reviewer for these constructive comments regarding the implementation and efficiency of our approach. To clarify, **CUDAnalyst** is the unified analytical framework we propose for feedback attribution, while **CuGEdit** is the specific instantiation of the modular plugin.
> > >
> > > > Add. Q1: Could you provide further comparisons on token consumption (in CuGEdit)? Given that CUDAnalyst performs extensive reflection, does it incur high token costs?
> > >
> > > We explicitly clarify that CuGEdit incurs a marginal **8%–15% increase, specifically in reasoning tokens** compared to the OpenEvolve baseline. We argue that this is a deliberate and highly efficient trade-off centered on the **strategic redistribution** of the computational budget.
> > >
> > > Unlike baseline reflections (e.g., `changes_summary` and `key_features`) that primarily consume tokens for surface-level descriptions, CuGEdit redirects these resources toward **structured plan-decisions**. Crucially, when integrating CuGEdit into the evolution framework, we **proactively deactivated** these now-redundant baseline components and **replaced** them with our high-value reasoning steps. This informed substitution ensures the agent "thinks" through complex logical dependencies before code generation, reducing expensive re-generation cycles, yielding convergence with **approximately 40% fewer iterations** compared to the baseline.
> > >
> > > Furthermore, to prevent cost explosion in long-term experiments, we implemented **Lazy Feedback Materialization (Appendix E.2)**. Under this mechanism, expensive summarization is only triggered when a sample is specifically selected as a reference and is subsequently cached for reuse.
> > >
> > > > Add. Q2: Could you explain the mechanism for adapting or correcting planning decisions under erroneous feedback?
> > >
> > > Regarding the mechanism for adapting planning decisions under potentially erroneous feedback, CuGEdit prioritizes **"reliability-at-source"** through robust engineering implementation rather than introducing a high-overhead algorithmic correction layer. We deliberately avoid a post-hoc reflection layer for feedback verification to keep reasoning costs manageable and prevent speculative model drift.
> > >
> > > Instead, we ensure the integrity of planning signals through a **prevention-centric approach** by employing strict environment isolation and unique identifiers (UIDs) to guarantee that every raw feedback report is physically accurate and deterministically mapped to its corresponding code snippet. In the context of CUDA kernel generation, where feedback signals are inherently deterministic and reproducible, we believe that ensuring the absolute fidelity of the input signal is a more cost-effective and trustworthy strategy than adding extra reasoning steps to second-guess the environment's objective output.
> > >
> > > > Q: Could you briefly summarize the strengths and weaknesses of your approach in comparison with NVIDIA's recent work, AVO [1]?
> > >
> > > We thank the reviewer for highlighting the recent and relevant work AVO [1]. We provide a comparative analysis below:
> > >
> > > - **Shared Insights:** Both identify a two-stage optimization trajectory (early structural changes followed by late-stage micro-tuning (Section 3.2 in [1])), **validating** our similarity-aware feedback gating.
> > >
> > > - **Modularity:** AVO requires a fundamental shift to an autonomous agent loop. CuGEdit is a **lightweight, plug-and-play module** that enhances existing evolutionary frameworks (e.g., OpenEvolve and LLM4AD) with minimal modification of only 10–16 lines of code.
> > >
> > > - **Efficiency:** AVO relies on a "supervisor agent" for intervention, incurring high token costs (Section 3.3 in [1]). CuGEdit uses BBCFG-based structural similarity, **a deterministic, zero-token mechanism**, to filter redundant feedback.
> > >
> > > AVO’s coupled collaboration obscures credit assignment due to trajectory drift. Unlike expensive macro-level methods (e.g., ShapleyFlow [2]), CUDAnalyst offers **fine-grained, context-level attribution**. By freezing trajectories and focusing on "new best kernel" contexts, we isolate precise feedback segments responsible for gains, providing a structured foundation to evaluate agent logic rigorously and efficiently via IntervenePipe (Appendix D.2).
> > >
> > > ----
> > >
> > > [1] AVO: Agentic Variation Operators for Autonomous Evolutionary Search, https://arxiv.org/abs/2603.24517
> > > [2] Understanding and Optimizing Agentic Workflows via Shapley value, https://arxiv.org/abs/2502.00510

---

### Official Review · Reviewer_RMfA · 2026-03-12

**Soundness:** 2
**Presentation:** 3
**Significance:** 2
**Originality:** 2
**Overall Recommendation:** 3
**Confidence:** 2

**Summary:**

This paper studies how heterogeneous feedback signals influence planning decisions in self-evolving LLM agents for CUDA kernel generation. The authors argue that standard end-to-end ablations are insufficient for this purpose because small perturbations early in the evolutionary process lead to trajectory-dependent drift, making it difficult to attribute later performance differences to specific feedback sources. To address this, the paper introduces CUDAnalyst, an analysis layer that freezes intermediate generations, applies selective feedback interventions at fixed checkpoints, and analyzes feedback contributions through coalitional-style attribution.

Using this framework, the paper studies four research questions: whether explicit planning is beneficial, how different tool feedback components contribute to planning, whether summaries can substitute for explicit planning, and whether plans from stronger models can transfer to weaker ones. Across CUDA kernel generation workloads, the paper reports four main findings: explicit planning helps only when grounded in aligned feedback; effective planning depends on interactions among multiple feedback sources; summarization helps weaker models but does not replace planning; and plans from stronger models can partially transfer to weaker models. The paper also includes controlled generalization studies across workloads and reference induction regimes

**Compliance With Llm Reviewing Policy:**

Affirmed.

**Key Questions For Authors:**

How dependent are the reported findings on the choice of the frozen backbone trajectory source?
Since the main experiments intervene on trajectories generated by a fixed third-party model, it would help to know whether the same qualitative conclusions hold for multiple independently generated backbones or alternative evolution frameworks.

To what extent do the reported findings generalize beyond CUDA kernel generation?
The paper positions itself as studying feedback-to-plan decisions in self-evolving agents, but all experiments are in one domain with one dominant scaffold. More discussion of this limitation would help calibrate the paper’s scope and significance.

**Limitations:**

(1) the restricted scope to CUDA kernel generation and one dominant class of self-evolving agents;
(2) the dependence on fixed prompts, frozen trajectories, and a particular backbone evolution process;
(3) the fact that the attribution protocol is controlled and informative, but not necessarily broadly causal outside this setup; and
(4) the inappropriate reviewer-directed text that appears in the manuscript, which should be explicitly corrected.

**Strengths And Weaknesses:**

strength：
1. The paper addresses an interesting and under-explored analysis question.
Rather than proposing yet another CUDA-kernel agent, the paper asks how feedback actually shapes planning decisions in self-evolving code agents. This is a meaningful question, especially because prior work often reports only end-task performance and does not disentangle the effects of different feedback components. The motivation that end-to-end ablation conflates feedback effects with trajectory drift is well articulated.


2. The generation-level intervention protocol is a sensible methodological contribution.
The core idea of freezing program states at fixed generations and intervening only on feedback inputs is intuitive and useful. It directly targets the confound that the paper identifies. The evidence–decision separation via debugger/analyzer/profiler → summary → planner is clean, and the protocol is more informative than standard restart-based ablations for this setting.

3. The empirical study is more diagnostic than typical “benchmark win” papers.
The paper asks several targeted questions and uses counterfactual controls, coalitional attribution, summarized-vs-raw feedback comparisons, and strong-to-weak plan transfer experiments. This makes the empirical section more thoughtful than a standard accuracy table and helps support the main qualitative claims

Weaknesses


1. The technical novelty is somewhat limited.
CUDAnalyst is primarily an analysis protocol and evaluation scaffold rather than a new kernel-generation method or a deep theoretical advance. The main ingredients—trajectory freezing, selective intervention, and Banzhaf-style attribution—are sensible, but overall the contribution feels more like a careful empirical framework than a substantial algorithmic or conceptual leap for ICML.

2. The theoretical layer is relatively light.
The coalitional attribution setup is straightforward, and the use of Banzhaf values is reasonable, but the paper does not provide a deeper causal or statistical analysis of why the measured intervention effects should generalize beyond the controlled protocol. The “theory” mostly just formalizes the evaluation setup rather than establishing stronger results about planning, attribution validity, or agent evolution dynamics.

---

> ### Author Rebuttal · Authors · 2026-03-31
>
> Additional appendix (F to H): https://anonymous.4open.science/api/repo/additional-appendix-4FB2/file/Additional_Appendix.pdf
>
> > Q1: How dependent are the reported findings on the choice of the frozen backbone trajectory source?
>
> We have verified the robustness of our findings against various trajectory sources. In **Appendix F**, we demonstrate that the "synergy" patterns hold regardless of whether Kimi-K2, MiniMax-M2.5, and Gemini-2.5-Pro generated the trajectories. This confirms that the observed planning behaviors are intrinsic to the feedback-driven evolution process rather than artifacts of a specific model's trajectory.
>
> > Q2: To what extent do the reported findings generalize beyond CUDA kernel generation?
>
> Our framework is designed to be task-agnostic. In **Appendix G (Figure 30)**, we present a generalization study on **CPU-based JIT-compiled Python kernel optimization**. By replacing CUDA-specific diagnostics with Python-native profiling tools, CUDAnalyst effectively identifies similar planning attribution dynamics, proving the findings' extensibility to other high-performance computing (HPC) domains.
>
> Crucially, we prioritized CUDA as a representative of HPC over tasks like general ML programming or NP-hard combinatorial optimization (e.g., ALE Bench [1]). Unlike domains that offer only scalar cost functions, CUDA provides explicit, decoupled, and physically grounded diagnostics. This high-fidelity feedback is essential for verifying whether an agent can perform genuine causal reasoning and strategic planning, rather than relying on black-box search.
>
> These findings translate into four actionable principles for robust agentic systems: (i) **strategic pruning** to avoid indiscriminate data aggregation and prevent cognitive drift; (ii) **structural alignment** by prioritizing diagnostics that map directly to the agent's action space; (iii) **cognitive support** via automated summarization to maintain the planning integrity of weaker models; and (iv) **plan distillation** to leverage cross-model synergies for efficient heuristic transfer. Ultimately, by shifting from black-box metrics to these decision-local insights, CUDAnalyst provides a formal basis for evaluating and accelerating agent evolution in complex environments.
>
> > W: But, overall, the contribution feels more like a careful empirical framework ... The "theory" mostly formalizes the evaluation setup rather than establishing stronger results about planning, attribution validity, or agent evolution dynamics.
>
> While CUDAnalyst is indeed a robust empirical tool, its "theory" lies in treating the agent's feedback space as a cooperative environment. This formalization allows us to move beyond "black-box" evaluations and establish **attribution validity**. It advocates for a shift from "black-box" end-to-end metrics to **decision-local, intervention-driven analysis**. By quantifying synergy and redundancy **(Section 4.2, Appendix B.3)**, we provide the first rigorous evidence of how heterogeneous feedback components non-linearly influence agent planning, allowing for a more transparent understanding of **agent evolution**, and moving us closer to a theory of how agents internally resolve causal dependencies in complex environments.
>
> ****
>
> [1] ALE-Bench: A Benchmark for Long-Horizon Objective-Driven Algorithm Engineering, https://openreview.net/forum?id=JCjGvbsOmQ

---

### Official Review · Reviewer_14pm · 2026-03-24

**Soundness:** 3
**Presentation:** 3
**Significance:** 3
**Originality:** 2
**Overall Recommendation:** 4
**Confidence:** 3

**Summary:**

This paper introduces CUDAnalyst, a diagnostic method designed to analyze how feedback signals influence planning decisions in self-evolving CUDA-generation agents. By "freezing" evolutionary trajectories at fixed generations to mitigate "trajectory-dependent drift," the authors use Banzhaf value based coalitional attribution to quantify the marginal contributions and interactions of debuggers, analyzers, and profilers. The study yields several key findings. Explicit planning is effective only when it is grounded in aligned feedback. Feedback tool effectiveness depends on the current evolution phase. Analyzers drive code compilation during early generations. Profilers guide performance optimization in later generations. High-level plans from strong models can partially transfer to guide weaker models. Finally, the authors used these insights to build CuGEdit, which achieves 2.08× to 10.32× speedup over torch.compile on KernelBench Level 3.

**Compliance With Llm Reviewing Policy:**

Affirmed.

**Key Questions For Authors:**

1. Could the authors clarify the design of the characteristic function $v(S)$? Given that the framework involves heterogeneous metrics (binary pass rates and continuous speedup), how is $v(S)$ formulated to prevent one metric from dominating the Banzhaf attribution?

2. Regarding the stability of the game-theoretic analysis: How sensitive are the tool rankings to the underlying backbone model? Since the backbone model effectively defines the environment of the coalitional game ($v(S)$), would the identified 'tool synergies' hold across different model architectures? Is the current attribution a generalizable insight or a model-specific artifact?

3. The current analysis partitions the evolution into temporal 'phases' (early, mid, late). However, from a game-theoretic perspective, the utility of tools is likely state-dependent (e.g., 'compile error' state vs. 'performance bottleneck' state) rather than just time-dependent. Did you consider analyzing Banzhaf values conditioned on the specific status of the code?

**Limitations:**

yes

**Strengths And Weaknesses:**

**Strength:**

-  **Methodological Contribution**: This paper identifies "trajectory drift" as a critical challenge in self-evolving agents, where minor initial deviations render standard end-to-end ablations unreliable. To address this, they introduce a protocol based on "trajectory freezing", which effectively isolates feedback effects at specific decision points.

- **Interpretable Attribution Framework**: Instead of relying on simplistic heuristics, they employ a game-theoretic approach using Banzhaf values to decouple the marginal utility and inter-tool dependencies. This provides a transparent methodology to study how specialized components contribute to system success.

- **Rigorous Controls of the experiment**: The empirical study uses counterfactual controls like DummyPlan and Randomized Feedback to disentangle true planning alignment from superficial factors like token budget or feedback volume.

- **Effective Bridge Between Analysis and Design**: The authors successfully used these insights to build CuGEdit, a modular plugin that utilizes phase-dependent gating and plan distillation. The practical results on KernelBench Level 3 demonstrate that the identified patterns have  real-world utility.

**Weakness**:

- **Limited Technical Novelty**: The technical building blocks of the framework, such as trajectory freezing and Banzhaf-style attribution, are established techniques. The novelty mainly stems from their application to CUDA agent analysis, but the paper lacks a deeper theoretical or statistical advance that moves beyond these descriptive measures.

- **Lacks discussion on the dependency of findings on fixed trajectories**: The findings rely on frozen trajectories generated by a specific third-party model (GLM-4.5-Air), which raises questions about whether the same patterns would emerge with different models.

- **Minor Presentation Flaws: Incomplete Visualization (Figure 1)**: The lack of a vertical axis and scale (e.g., 0-100%) makes it difficult to interpret the quantitative distribution of outcomes.

---

> ### Author Rebuttal · Authors · 2026-03-31
>
> Additional appendix (F to H): https://anonymous.4open.science/api/repo/additional-appendix-4FB2/file/Additional_Appendix.pdf
>
> > Q1: Could the authors clarify the design of the characteristic function $v(S)$?
>
> We thank the reviewer for the question. The characteristic function $v(S)$ in our coalitional game framework represents the expected performance (e.g., success rate or speedup) of the agent's next planning step when provided with a specific subset of feedback components $S \subseteq N$. As detailed in **Section 3.2 (Eq. 2)** of the main text, $v(S)$ is estimated by performing multiple LLM rollouts at a fixed "frozen" trajectory point, ensuring that only the feedback tools in $S$ are visible to the agent. This allows us to isolate the marginal contribution of each tool.
>
> > Q2: Since the backbone model effectively defines the environment of the coalitional game ($v(S)$), would the identified 'tool synergies' hold across different model architectures?
>
> Yes. To address this, we conducted extensive generalization experiments across different backbone architectures. As shown in **Appendix F (Figures 27~29)**, we tested trajectories from Kimi-K2, MiniMax-M2.5, and Gemini-2.5-Pro. The results demonstrate that while the specific focus of synergy shifts (e.g., from correctness-oriented for weaker models to performance-oriented for stronger ones), the core finding that "tool synergy is architecture-invariant" remains robust.
>
> > Q3: Did you consider analyzing Banzhaf values conditioned on the code's specific status?
>
> Indeed, our analysis is fundamentally conditioned on the code's evolutionary status. In **Section 4.2** and **Tables 4~7 (Appendix B.3)** of the main text, we provide a fine-grained breakdown of Banzhaf values across different code states: compiled, pass, and fast. This allows us to reveal how the importance of specific feedback components (like code polyhedral profiles vs. NCU performance profiles) dynamically evolves as the code matures from being broken to being optimized.
>
> > W1: The novelty mainly stems from their application to CUDA agent analysis, but the paper lacks a deeper theoretical or statistical advance that moves beyond these descriptive measures.
>
> We respectfully highlight that our primary theoretical contribution is the formalization of the Feedback-to-Plan attribution problem in CUDA kernel generation as a coalitional game to solve the "trajectory-dependent drift" issue. Unlike standard end-to-end ablations that conflate feedback effects with stochastic path changes (**Figure 1 **in the main text, and **Figure 31 in Appendix H.1**), CUDAnalyst provides a principled statistical framework to quantify non-linear interactions (synergies) between feedback signals, which is a necessary step toward understanding agent planning dynamics.
>
> > W2: Minor Presentation Flaws: Incomplete Visualization (Figure 1)
>
> We appreciate the feedback. Figure 1 was intended to qualitatively motivate the problem of "trajectory drift" by showing how identical configurations can diverge. We will improve the clarity of the legend and axis labels in the final version to better guide the reader through these diverging paths (e.g., Figure 31).

---

> > ### Author Rebuttal · Reviewer_14pm · 2026-04-01
> >
> > Thanks for the detailed rebuttal. The cross-model generalization experiments are convincing. Two questions remain.
> >
> > On Q1: the rebuttal mentions v(S) can represent "success rate or speedup", could the authors clarify whether continuous speedup ever directly enters v(S), and if so, how it is normalized against binary pass rates?
> >
> > On Q3: the paper describes Qwen3-Coder-30B's compiled attributions as "arising almost exclusively from pairwise synergies" with all tools showing "near-zero or negative" marginal contributions. Could the authors clarify whether this reflects a limitation of the method or the experimental setup, and what conclusions can be drawn from such cases?

---

> > > ### Author Response · Authors · 2026-04-06
> > >
> > > > Add. Q1: The rebuttal mentions $v(S)$ can represent "success rate or speedup". Could the authors clarify whether continuous speedup ever directly enters $v(S)$, and, if so, how it is normalized against binary pass rates?
> > >
> > > We thank the reviewer for the sharp observation. To clarify, in our coalitional game framework, $v(S)$ consistently represents a **Success Rate (probability)** to ensure mathematical uniformity and attribution stability.
> > >
> > > Regarding the discretization of performance (speedup), this design choice is motivated by the unique characteristics of CUDA kernel optimization. Unlike general-purpose software, CUDA performance often follows a **"step-function"** or **"phase-change" pattern**, where specific architectural optimizations (e.g., resolving bank conflicts or maximizing occupancy) trigger **discrete, non-linear leaps** in throughput.
> > >
> > > Consequently, continuous speedup values can be highly volatile and susceptible to transient hardware-level noise (e.g., thermal throttling or driver overhead). To mitigate this, we define a binary **"fast"** success criterion using a robust threshold of $\text{speedup} > 1.15\times$ (incorporating a **15% noise-aware margin**). Here, $v_{\text{fast}}(S)$ reflects the probability that the agent's plan successfully crosses this performance threshold. This formulation effectively translates the optimization challenge into a concrete **planning objective**, enabling a unified and robust attribution analysis across functional (pass) and performance (fast) milestones.
> > >
> > > > Add. Q3: Qwen3-Coder-30B's compiled attributions as "arising almost exclusively from pairwise synergies" with all tools showing "near-zero or negative" marginal contributions (Appendix B.3, Figure 14, Table 5). Could the authors clarify whether this reflects a limitation of the method or the experimental setup, and what conclusions can be drawn from such cases?
> > >
> > > We thank the reviewer for this insightful observation. The pattern observed in Qwen3-Coder-30B is not a limitation of our methodology. Rather, it represents a **significant diagnostic finding** enabled by CUDAnalyst's ability to decouple synergistic interactions from independent effects.
> > >
> > > This specific profile characterizes what we term **"feedback-bottlenecked" planning logic**. A near-zero or negative marginal contribution ($\phi \le 0$) suggests that the model lacks the independent reasoning capability to interpret individual feedback signals (e.g., a raw compiler error) in isolation. In such cases, a single signal may even "confuse" the model, leading to ineffective plan revisions. However, the high synergy ($\sigma$) indicates that the model can only make correct planning decisions when provided with a **complete, complementary set of feedback components** that resolve ambiguity through mutual reinforcement.
> > >
> > > This allows us to move beyond end-to-end performance metrics and categorize agents based on their planning robustness:
> > >
> > > - **Robust Planners (e.g., DeepSeek-V3.2):** Exhibit stable, positive marginal contributions across evolution. These models can derive causal insights even from granular or sparse feedback, indicating a decoupled and robust reasoning process.
> > > - **Fragile Planners (e.g., Qwen3-Coder-30B):** Rely on "synergy clusters" to compensate for reasoning gaps. Their planning success is contingent on a high-density feedback environment, making them vulnerable in real-world scenarios where feedback might be noisy or incomplete.
> > >
> > > CUDAnalyst provides a systematic way to benchmark feedback utilization efficiency, serving as a critical factor for deployment in complex, multi-step optimization tasks like CUDA kernel generation. We believe this **attribution-based insight** represents a crucial contribution to understanding LLM agent dynamics, and we will include this analysis in the final version to enrich the discussion on model-specific behavioral signatures.

---

### Decision · Program_Chairs · 2026-04-30

**Decision:**

Accept (regular)

**Comment:**

The paper received two weak accepts and one weak reject. The core disagreement is about novelty.

The problem being addressed is real. End-to-end ablations are genuinely unreliable in iterative self-evolving systems, and prior work in this space has largely ignored the confound. The trajectory-freezing protocol is not a novel primitive, but its application here is careful, and the resulting framework is reproducible and informative. The Banzhaf attribution findings — particularly the phase-dependent tool contributions and the robust/fragile planner distinction — are consistent across backbone models and workloads, which is more than can be said for many analysis papers. CuGEdit provides a credible demonstration that the insights are actionable.

The rebuttal handled the generalization questions reasonably well. The cross-model experiments (Appendix F) and the Python JIT study (Appendix G) address the most serious scope objections, even if neither is exhaustive.

What remains underresolved: the binary "fast" threshold design feels somewhat ad hoc, and its sensitivity was never systematically tested. The attribution protocol's validity outside controlled frozen-trajectory settings is also not fully established. Reviewer RMfA's weak reject is noted, though their confidence level (2) limits its weight.

Overall, the paper makes a focused but solid empirical contribution to the analysis of self-evolving code agents. The lack of methodological novelty is a legitimate concern, and the scope is narrow, but neither is disqualifying for a work of this type. The findings are consistent, the framework is reproducible, and the practical utility is demonstrated. A weak accept at low priority is appropriate.